# UniDB: A Unified Diffusion Bridge Framework via Stochastic Optimal Control

**Kaizhen Zhu** [1 2 *]  **Mokai Pan** [1 2 *]  **Yuexin Ma** [1 2]  **Yanwei Fu** [3]  **Jingyi Yu** [1 2]  **Jingya Wang** [1 2]  **Ye Shi** [1 2 †]

## Abstract

Recent advances in diffusion bridge models leverage Doob's $h$-transform to establish fixed endpoints between distributions, demonstrating promising results in image translation and restoration tasks. However, these approaches frequently produce blurred or excessively smoothed image details and lack a comprehensive theoretical foundation to explain these shortcomings. To address these limitations, we propose UniDB, a unified framework for diffusion bridges based on Stochastic Optimal Control (SOC). UniDB formulates the problem through an SOC-based optimization and derives a closed-form solution for the optimal controller, thereby unifying and generalizing existing diffusion bridge models. We demonstrate that existing diffusion bridges employing Doob's $h$-transform constitute a special case of our framework, emerging when the terminal penalty coefficient in the SOC cost function tends to infinity. By incorporating a tunable terminal penalty coefficient, UniDB achieves an optimal balance between control costs and terminal penalties, substantially improving detail preservation and output quality. Notably, UniDB seamlessly integrates with existing diffusion bridge models, requiring only minimal code modifications. Extensive experiments across diverse image restoration tasks validate the superiority and adaptability of the proposed framework. Our code is available at https://github.com/UniDB-SOC/UniDB/.

## 1. Introduction

The diffusion model has been extensively utilized across a range of applications, including image generation and editing (Ho et al., 2020; Kawar et al., 2022; Song et al., 2020; Xia et al., 2023; Li et al., 2023), imitation learning (Wu et al., 2024; Chi et al., 2023; Ze et al., 2024) and reinforcement learning (Yang et al., 2023; Ding et al., 2024a), etc. Despite its versatility, the standard diffusion model faces limitations in transitioning between arbitrary distributions due to its inherent assumption of a Gaussian noise prior. To overcome this problem, diffusion models (Dhariwal & Nichol, 2021; Ho & Salimans, 2022; Murata et al., 2023; Ding et al., 2024b; Chung et al., 2022; Tang et al., 2024) often rely on meticulously designed conditioning mechanisms and classifier/loss guidance to facilitate conditional sampling and ensure output alignment with a target distribution. However, these methods can be cumbersome and may introduce manifold deviations during the sampling process. Meanwhile, Diffusion Schrödinger Bridge (Shi et al., 2024; De Bortoli et al., 2021; Somnath et al., 2023) involves constraints that hinder direct optimization of the KL divergence, resulting in slow convergence and limited model fitting capability.

To address this challenge, DDBMs (Zheng et al., 2024) proposed a diffusion bridge model using Doob's $h$-transform. This framework is specifically designed to establish fixed endpoints between two distinct distributions by learning the score function of the diffusion bridge from data, and then solving the stochastic differential equation (SDE) based on these learned scores to transition from one endpoint distribution to another. However, the forward SDE in DDBMs lacks the mean information of the terminal distribution, which restricts the quality of the generated images, particularly in image restoration tasks. Subsequently, GOUB (Yue et al., 2023) extends this framework by integrating Doob's $h$-transform with a mean-reverting SDE, achieving better results compared to DDBMs. Despite the promising results in diffusion bridge with Doob's $h$-transform, two fundamental challenges persist: 1) the theoretical mechanisms by which Doob's $h$-transform governs the bridging process remain poorly understood, lacking a rigorous framework to unify its empirical success; and 2) while effective for global distribution alignment, existing methods frequently degrade high-frequency details—such as sharp edges and fine textures—resulting in outputs with blurred or oversmoothed artifacts that compromise perceptual fidelity. These limitations underscore the need for both theoretical grounding and enhanced detail preservation in diffusion bridges.

---

[*]Equal contribution  [1]ShanghaiTech University  [2]MoE Key Laboratory of Intelligent Perception and Human-Machine Collaboration  [3]Fudan University.  Correspondence to: Ye Shi <shiye@shanghaitech.edu.cn>.

*Proceedings of the 42nd International Conference on Machine Learning*, Vancouver, Canada. PMLR 267, 2025. Copyright 2025 by the author(s).

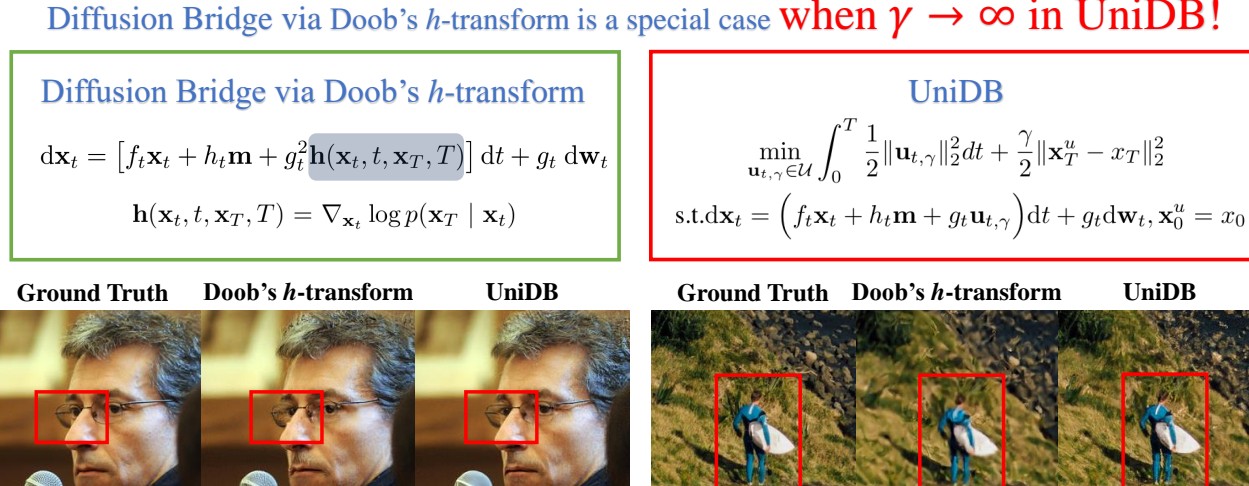

*Figure 1.* Recent advances in diffusion bridge models leverage Doob's $h$-transform to establish fixed endpoints between distributions, which introduces an $h$ function into the forward process of a standard stochastic differential equation (SDE) to forcibly match the two endpoints. However, as shown in the figure, this method can lead to local blurring and distortion in the generated images. UniDB formulates the forward process as a stochastic optimal control problem and employs the penalty coefficient $\gamma$, balancing realistic SDE trajectories and target endpoint matching, to produce images with more realistic details. We also find that Doob's $h$-transform is a special case in our framework when $\gamma \to \infty$. Therefore, our framework can seamlessly integrate with existing diffusion bridge models (with Doob's $h$-transform).

In this paper, we revisit the diffusion bridges through the lens of stochastic optimal control (SOC) by introducing a novel framework called UniDB, which formulates an optimization problem based on SOC principles to implement diffusion bridges. It enables the derivation of a closed-form solution for the optimal controller, along with the corresponding training objective for the diffusion bridge. UniDB identifies Doob's $h$-transform as a special case when the terminal penalty coefficient in the SOC cost function approaches infinity. This explains why Doob's $h$-transform may result in suboptimal solutions with blurred or distorted details. To address this limitation, UniDB utilizes the penalty coefficient in SOC to adjust the expressiveness of the image details and enhance the authenticity of the generated outputs. Our main contributions are as follows:

- We introduce UniDB, a novel unified diffusion bridge framework based on stochastic optimal control. This framework generalizes existing diffusion bridge models like DDBMs and GOUB, offering a comprehensive understanding and extension of Doob's $h$-transform by incorporating general forward SDE forms.

- We derive closed-form solutions for the SOC problem, demonstrating that Doob's $h$-transform is merely a special case within UniDB when the terminal penalty coefficient in the SOC cost function approaches infinity. This insight reveals inherent limitations in the existing diffusion bridge approaches, which UniDB overcomes. Notably, the improvement of UniDB requires minimal code modification, ensuring easy implementation.

- UniDB achieves state-of-the-art results in various image restoration tasks, including super-resolution (DIV2K), inpainting (CelebA-HQ), and deraining (Rain100H), which highlights the framework's superior image quality and adaptability across diverse scenarios.

## 2. Related Work

**Diffusion with Guidance.** This technique tackles conditional generative tasks by leveraging a differentiable loss function for guidance without the need for additional training (Chung et al., 2022; Shenoy et al., 2024; Bradley & Nakkiran, 2024). However, it often yields suboptimal image quality and a prolonged sampling process due to the

necessity of small step sizes. Most importantly, the sampling process is prone to manifold deviations and detail losses (Yang et al., 2024). Furthermore, enhancing the guidance of the diffusion typically requires the introduction of additional modules, thereby increasing the model's computational complexity.

**Diffusion Schrödinger Bridge.** This approach aims to determine a stochastic process, $\pi^*$ that facilitates probabilistic transport between a given initial distribution $P_{\text{prior}}$, and a terminal distribution $P_{\text{data}}$ (Liu et al., 2023; Shi et al., 2024; De Bortoli et al., 2021) while minimizing the Kullback-Leibler (KL) divergence. However, its training process is usually intricate, involving constraints that hinder direct optimization of the KL divergence, resulting in slow convergence and limited model fitting capability. For instance, DSB (Somnath et al., 2023) requires two independent forward passes during training to obtain the target distribution, thereby increasing both the complexity and time cost of training.

**Diffusion Bridge with Doob's $h$-transform.** Recent advances in diffusion bridging have demonstrated the efficacy of Doob's $h$-transform in enhancing transition quality between arbitrary distributions. Notably, DDBMs (Zhou et al., 2023) pioneered this approach by employing a linear SDE combined with Doob's $h$-transform to construct direct diffusion bridges. Subsequently, GOUB (Yue et al., 2023) extends this framework by integrating Doob's $h$-transform with a mean-reverting SDE, achieving state-of-the-art performance in image restoration tasks. Despite these empirical successes, the theoretical foundations of Doob's $h$-transform in this context remain insufficiently explored. In addition, these methods often result in images with blurred or oversmoothed features, particularly affecting the capture of high-frequency details crucial for perceptual fidelity.

**Diffusion with Stochastic Optimal Control.** The integration of SOC principles into diffusion models has emerged as a promising paradigm for guiding distribution transitions. DIS (Berner et al., 2022) established a foundational theoretical linkage between diffusion processes and SOC, while RB-Modulation (Rout et al., 2024) operationalized SOC via a simplified SDE structure for training-free style transfer using pre-trained diffusion models. Close to our work, DBFS (Park et al., 2024) leveraged SOC to construct diffusion bridges in infinite-dimensional function spaces and also established equivalence between SOC and Doob's $h$-transform. However, DBFS primarily extends Doob's $h$-transform to infinite Hilbert spaces via SOC, without addressing its intrinsic limitations. Our analysis reveals a critical insight: Doob's $h$-transform corresponds to a suboptimal solution that can inherently lead to artifacts such as blurred or distorted details. To resolve this, we introduce a unified SOC framework that jointly optimizes trajectory costs and terminal constraints, enhancing detail preservation and image quality.

## 3. Preliminaries

### 3.1. Denoising Diffusion Bridge Models

Starting with an initial $d$-dimensional data distribution $\mathbf{x}_0 \sim q_{\text{data}}(\mathbf{x})$, diffusion models (Song et al., 2020; Ho et al., 2020; Sohl-Dickstein et al., 2015; Song & Ermon, 2019) construct a diffusion process, which can be achieved by defining a forward stochastic process evolving from $\mathbf{x}_0$ through a stochastic differential equation (SDE):

$$\mathrm{d}\mathbf{x}_t = \mathbf{f}(\mathbf{x}_t, t)\mathrm{d}t + g_t\,\mathrm{d}\mathbf{w}_t, \tag{1}$$

where $t$ ranges over the interval $[0, T]$, $\mathbf{f} : \mathbb{R}^d \times [0, T] \to \mathbb{R}^d$ is the vector-valued drift function, $g : [0, T] \to \mathbb{R}$ signifies the scalar-valued diffusion coefficient and $\mathbf{w}_t \in \mathbb{R}^d$ is the Wiener process, also known as Brownian motion. To promise the transition probability $p(\mathbf{x}_t \mid \mathbf{x}_s)$ remains Gaussian, almost all the diffusion SDEs take the following linear form (Zheng et al., 2024) in (1):

$$\mathbf{f}(\mathbf{x}_t, t) = f(t)\mathbf{x}_t, \tag{2}$$

where $f(t)$ is some scalar-valued function. To realize transition between arbitrary distributions, DDBMs introduces Doob's $h$-transform (Särkkä & Solin, 2019), a mathematical technique applied to stochastic processes, which rectifies the drift term of the forward diffusion process to pass through a preset terminal point $\mathbf{x}_T \in \mathbb{R}^d$. Precisely, the forward process of diffusion bridges after Doob's $h$-transform becomes:

$$\mathrm{d}\mathbf{x}_t = \left[\mathbf{f}(\mathbf{x}_t, t) + g_t^2 \mathbf{h}(\mathbf{x}_t, t, \mathbf{x}_T, T)\right]\mathrm{d}t + g_t\,\mathrm{d}\mathbf{w}_t, \tag{3}$$

where $\mathbf{h}(\mathbf{x}_t, t, \mathbf{x}_T, T) = \nabla_{\mathbf{x}_t} \log p(\mathbf{x}_T \mid \mathbf{x}_t)$ is the $h$ function. The diffusion bridge can connect the initial $\mathbf{x}_0$ to any given terminal $\mathbf{x}_T$ and thus is promising for various image restoration tasks. Meanwhile, its backward reverse SDE (Anderson, 1982) is given by

$$\begin{aligned}
\mathrm{d}\mathbf{x}_t = \Big[&\mathbf{f}(\mathbf{x}_t, t) + g_t^2 \nabla_{\mathbf{x}_t} \log p(\mathbf{x}_T \mid \mathbf{x}_t) \\
&- g_t^2 \nabla_{\mathbf{x}_t} \log p(\mathbf{x}_t \mid \mathbf{x}_T)\Big]\mathrm{d}t + g_t\,\mathrm{d}\tilde{\mathbf{w}}_t.
\end{aligned} \tag{4}$$

where $\tilde{\mathbf{w}}_t$ is the reverse-time Wiener process and the unknown term $\nabla_{\mathbf{x}_t} \log p(\mathbf{x}_t \mid \mathbf{x}_T)$ can be estimated by a score prediction neural network $s_\theta$ (Song et al., 2020).

### 3.2. Generalized Ornstein-Uhlenbeck Bridge

Generalized Ornstein-Uhlenbeck (GOU) process describes a mean-reverting stochastic process commonly used in finance, physics, and other fields in the following SDE form (Ahmad, 1988; Pavliotis & Pavliotis, 2014; Wang et al., 2018):

$$\mathrm{d}\mathbf{x}_t = \theta_t\left(\boldsymbol{\mu} - \mathbf{x}_t\right)\mathrm{d}t + g_t\,\mathrm{d}\mathbf{w}_t, \tag{5}$$

where $\boldsymbol{\mu}$ is a given state vector, $\theta_t$ denotes a scalar drift coefficient and $g_t$ represents the diffusion coefficient with $\theta_t$, $g_t$ satisfying the specified relationship $g_t^2 = 2\lambda^2\theta_t$ where $\lambda^2$ is a given constant scalar. Based on this, Generalized Ornstein-Uhlenbeck Bridge (GOUB) is a diffusion bridge model (Yue et al., 2023), which can address image restoration tasks without the need for specific prior knowledge if we consider the initial state $\mathbf{x}_0$ to represent a high-quality image and the corresponding low-quality image $\mathbf{x}_T = \boldsymbol{\mu}$ as the final condition. With the introduction of $\boldsymbol{\mu}$, $\mathbf{x}_t$ tends to $\boldsymbol{\mu}$ as time $t$ progresses. Through Doob's $h$-transform, denote $\bar{\theta}_{s:t} = \int_s^t \theta_z dz$, $\bar{\theta}_t = \int_0^t \theta_z dz$ for simplification when $s = 0$ and $\bar{\sigma}_{s:t}^2 = \lambda^2(1 - e^{-2\bar{\theta}_{s:t}})$, the forward process of GOUB is formed as:

$$\mathrm{d}\mathbf{x}_t = \left(\theta_t + g_t^2 \frac{e^{-2\bar{\theta}_{t:T}}}{\bar{\sigma}_{t:T}^2}\right)(\mathbf{x}_T - \mathbf{x}_t)\,\mathrm{d}t + g_t\,\mathrm{d}\mathbf{w}_t. \quad (6)$$

And the forward transition $p(\mathbf{x}_t \mid \mathbf{x}_0, \mathbf{x}_T)$ is given by

$$p(\mathbf{x}_t \mid \mathbf{x}_0, \mathbf{x}_T) = \mathcal{N}(\bar{\boldsymbol{\mu}}_t', \bar{\sigma}_t'^2\mathbf{I}),$$
$$\bar{\boldsymbol{\mu}}_t' = e^{-\bar{\theta}_t}\frac{\bar{\sigma}_{t:T}^2}{\bar{\sigma}_T^2}\mathbf{x}_0 + (1 - e^{-\bar{\theta}_t}\frac{\bar{\sigma}_{t:T}^2}{\bar{\sigma}_T^2})\mathbf{x}_T, \ \bar{\sigma}_t'^2 = \frac{\bar{\sigma}_t^2\bar{\sigma}_{t:T}^2}{\bar{\sigma}_T^2}. \quad (7)$$

Also, GOUB presents a new reverse ODE called Mean-ODE, which directly neglects the Brownian term of (4):

$$\mathrm{d}\mathbf{x}_t = \Big[\mathbf{f}(\mathbf{x}_t, t) + g_t^2\nabla_{\mathbf{x}_t}\log p(\mathbf{x}_T \mid \mathbf{x}_t) \\ - g_t^2\nabla_{\mathbf{x}_t}\log p(\mathbf{x}_t \mid \mathbf{x}_T)\Big]\mathrm{d}t. \quad (8)$$

### 3.3. Stochastic Optimal Control

Stochastic Optimal Control (SOC) is a mathematical discipline that focuses on determining optimal control strategies for dynamic systems under uncertainty. By integrating stochastic processes with optimization theory, SOC seeks to identify the best control strategies in scenarios involving randomness, as commonly encountered in fields like finance (Geering et al., 2010) and style transfer (Rout et al., 2024). Considering the dynamics described in (1), let us examine the following Linear Quadratic SOC problem (Bryson, 2018; O'Connell, 2003; Kappen, 2008; Chen et al., 2023):

$$\min_{\mathbf{u}_{t,\gamma}\in\mathcal{U}} \mathbb{E}\left[\int_0^T \frac{1}{2}\|\mathbf{u}_{t,\gamma}\|_2^2\,dt + \frac{\gamma}{2}\|\mathbf{x}_T^u - x_T\|_2^2\right] \quad (9)$$
$$\text{s.t. } \mathrm{d}\mathbf{x}_t = (\mathbf{f}(\mathbf{x}_t, t) + g_t\mathbf{u}_{t,\gamma})\,\mathrm{d}t + g_t\,\mathrm{d}\mathbf{w}_t, \ \mathbf{x}_0^u = x_0,$$

where $\mathbf{x}_t^u$ is the diffusion process under control, $x_0$ and $x_T$ represent for the initial state and the preset terminal respectively, $\|\mathbf{u}_{t,\gamma}\|_2^2$ is the instantaneous cost, $\frac{\gamma}{2}\|\mathbf{x}_T^u - x_T\|_2^2$ is the terminal cost with its penalty coefficient $\gamma$. The SOC problem aims to design the controller $\mathbf{u}_{t,\gamma}$ to drive the dynamic system from $x_0$ to $x_T$ with minimum cost.

## 4. Methods

### 4.1. Diffusion Bridges Constructed by SOC Problem

The forward SDE of the Diffusion Bridge with Doob's $h$-transform is enforced to pass from the predetermined origin $x_0$ to the terminal $x_T$. With a similar purpose, UniDB constructs a SOC problem where the constraints are an arbitrary linear SDE of the forward diffusion with a given initial state, while the objective incorporates a penalty term steering the forward diffusion trajectory towards the predetermined terminal $x_T$. Meanwhile, compared with the linear drift term (2), we combined a given state vector term $\mathbf{m}$ with the same dimension as $\mathbf{x}_t$ and its related coefficient $h_t$ which is a simple reformulation and generalization of the parameters in GOU process (5):

$$\mathbf{f}(\mathbf{x}_t, t) = f_t\mathbf{x}_t + h_t\mathbf{m}. \quad (10)$$

Accordingly, our SOC problem with unified linear SDE (10) is formed as:

$$\min_{\mathbf{u}_{t,\gamma}\in\mathcal{U}} \mathbb{E}\left[\int_0^T \frac{1}{2}\|\mathbf{u}_{t,\gamma}\|_2^2\,dt + \frac{\gamma}{2}\|\mathbf{x}_T^u - x_T\|_2^2\right]$$
$$\text{s.t.}\mathrm{d}\mathbf{x}_t = \Big(f_t\mathbf{x}_t + h_t\mathbf{m} + g_t\mathbf{u}_{t,\gamma}\Big)\mathrm{d}t + g_t\mathrm{d}\mathbf{w}_t, \mathbf{x}_0^u = x_0. \quad (11)$$

According to the certainty equivalence principle (Chen et al., 2023; Rout et al., 2024), the addition of noise or perturbations to a linear system with quadratic costs does not change the optimal control. Therefore, we can modify the SOC problem with the deterministic ODE condition to obtain the optimal controller $\mathbf{u}_{t,\gamma}^*$ as follows,

$$\min_{\mathbf{u}_{t,\gamma}\in\mathcal{U}} \int_0^T \frac{1}{2}\|\mathbf{u}_{t,\gamma}\|_2^2\,dt + \frac{\gamma}{2}\|\mathbf{x}_T^u - x_T\|_2^2 \quad (12)$$
$$\text{s.t. } \mathrm{d}\mathbf{x}_t = \Big(f_t\mathbf{x}_t + h_t\mathbf{m} + g_t\mathbf{u}_{t,\gamma}\Big)\mathrm{d}t, \ \mathbf{x}_0^u = x_0.$$

We can derive the closed-form solution to the problem (12), which leads to the following Theorem 4.1:

**Theorem 4.1.** *Consider the SOC problem* (12), *denote* $d_{t,\gamma} = \gamma^{-1} + e^{2\bar{f}_T}\bar{g}_{t:T}^2$, $\bar{f}_{s:t} = \int_s^t f_z dz$, $\bar{h}_{s:t} = \int_s^t e^{-\bar{f}_z}h_z dz$ *and* $\bar{g}_{s:t}^2 = \int_s^t e^{-2\bar{f}_z}g_z^2 dz$, *denote* $\bar{f}_t$, $\bar{h}_t$ *and* $\bar{g}_t^2$ *for simplification when* $s = 0$, *then the closed-form optimal controller* $\mathbf{u}_{t,\gamma}^*$ *is*

$$\mathbf{u}_{t,\gamma}^* = g_t e^{\bar{f}_{t:T}}\frac{x_T - e^{\bar{f}_{t:T}}\mathbf{x}_t - \mathbf{m}e^{\bar{f}_T}\bar{h}_{t:T}}{d_{t,\gamma}}, \quad (13)$$

*and the transition of* $\mathbf{x}_t$ *from* $x_0$ *and* $x_T$ *is*

$$\mathbf{x}_t = e^{\bar{f}_t}\left(\frac{d_{t,\gamma}}{d_{0,\gamma}}x_0 + \frac{e^{\bar{f}_T}\bar{g}_t^2}{d_{0,\gamma}}x_T + \Big(\bar{h}_t - \frac{e^{2\bar{f}_T}\bar{h}_T\bar{g}_t^2}{d_{0,\gamma}}\Big)\mathbf{m}\right). \quad (14)$$

The proof of Theorem 4.1 is provided in Appendix A.1. With Theorem 4.1, we can obtain an optimally controlled forward SDE connected from $x_0$ to the neighborhood of the terminal $x_T$ and the transition of $\mathbf{x}_t$ for the forward process. As for the backward process, similar to (4) and (8), the backward reverse SDE and Mean-ODE are respectively formulated as:

$$
\mathrm{d}\mathbf{x}_t = \Big[ f_t\mathbf{x}_t + h_t\mathbf{m} + g_t\mathbf{u}_{t,\gamma}^* 
$$
$$
\qquad - g_t^2 \nabla_{\mathbf{x}_t} \log p(\mathbf{x}_t \mid x_T)\Big]\mathrm{d}t + g_t\mathrm{d}\tilde{\mathbf{w}}_t,
$$
$$
\tag{15}
$$
$$
\mathrm{d}\mathbf{x}_t = \Big[ f_t\mathbf{x}_t + h_t\mathbf{m} + g_t\mathbf{u}_{t,\gamma}^* - g_t^2 \nabla_{\mathbf{x}_t} \log p(\mathbf{x}_t \mid x_T)\Big]\mathrm{d}t.
$$
$$
\tag{16}
$$

### 4.2. Connections between SOC and Doob's $h$-transform

We can intuitively see from the SOC problem that when $\gamma \to \infty$ in Theorem 4.1, it means that the target of SDE process is precisely the predetermined endpoint (Chen et al., 2023), which is also the purpose of Doob's $h$-transform and facilitates the following theorem:

**Theorem 4.2.** *For the SOC problem* (12)*, when $\gamma \to \infty$, the optimal controller becomes $\mathbf{u}_{t,\infty}^* = g_t \nabla_{\mathbf{x}_t} \log p(\mathbf{x}_T \mid \mathbf{x}_t)$, and the corresponding forward and backward SDE with the linear SDE form* (10) *are the same as Doob's $h$-transform as in* (3) *and* (4)*.*

The proof of Theorem 4.2 is presented in Appendix A.2. This theorem shows that existing diffusion bridge models using Doob's $h$-transform are merely special instances of our UniDB framework, which offers a unified approach to diffusion bridges through the lens of SOC.

Furthermore, using Doob's $h$-transform in diffusion bridge models is not necessarily optimal, as letting the terminal penalty coefficient $\gamma \to \infty$ eliminates the consideration of control costs in SOC. To support this argument, we present Proposition 4.3, which asserts that the diffusion bridge with Doob's $h$-transform is not the most effective choice.

**Proposition 4.3.** *Consider the SOC problem* (12)*, denote $\mathcal{J}(\mathbf{u}_{t,\gamma}, \gamma) \triangleq \int_0^T \frac{1}{2} \|\mathbf{u}_{t,\gamma}\|_2^2 \, dt + \frac{\gamma}{2} \|\mathbf{x}_T^u - x_T\|_2^2$ as the overall cost of the system, $\mathbf{u}_{t,\gamma}^*$ as the optimal controller* (13)*, then*

$$
\mathcal{J}(\mathbf{u}_{t,\gamma}^*, \gamma) \leq \mathcal{J}(\mathbf{u}_{t,\infty}^*, \infty). \tag{17}
$$

Detailed proof of Proposition 4.3 is provided in Appendix A.3. Proposition 4.3 shows that finite $\gamma$ achieves a lower total cost not by sacrificing performance, but by optimally trading minor terminal mismatches for significantly smoother and more natural diffusion paths. Doob's $h$-transform requires larger controller $\|\mathbf{u}_{t,\infty}^*\|_2^2 \geq \|\mathbf{u}_{t,\gamma}^*\|_2^2$ in SDE trajectory to force exact endpoint matching (the controlled target is precisely the preset endpoint $\|\mathbf{x}_T^{u^*} - x_T\|_2^2 = 0$ when

$\gamma \to \infty$), which may disrupt the inherent continuity and smoothness of images. Prioritizing pixel-perfect endpoints over smooth trajectories leads to "mathematically correct but visually unrealistic" outputs. As shown in Figure 1, Doob's $h$-transform can lead to artifacts along edges and unnatural patterns in smooth regions. Therefore, maintaining the penalty coefficient $\gamma$ as a hyperparameter is a more effective approach.

### 4.3. Training objective of UniDB

In this section, we focus on constructing the training objective of UniDB. According to maximum log-likelihood (Ho et al., 2020) and conditional score matching (Song et al., 2020), the training objective is based on the forward transition $p(\mathbf{x}_t \mid \mathbf{x}_0, \mathbf{x}_T)$. Thus, we begin by deriving this probability. The closed-form expression in (14) represents the mean value of the forward transition after applying reparameterization techniques. However, this expression lacks a noise component after the transformation based on the certainty equivalence principle. To address this issue, we employ stochastic interpolant theory (Albergo et al., 2023) to introduce a noise term $\bar{\sigma}_t'\epsilon$ with $\bar{\sigma}_0' = \bar{\sigma}_T' = 0$. We define $\bar{\sigma}_t'^2 = \bar{\sigma}_t^2 \bar{\sigma}_{t:T}^2 / \bar{\sigma}_T^2$ similar to (7), leading to the following forward transition:

$$
p(\mathbf{x}_t \mid x_0, x_T) = \mathcal{N}(\bar{\boldsymbol{\mu}}_{t,\gamma}, \bar{\sigma}_t'^2\mathbf{I}),
$$
$$
\bar{\boldsymbol{\mu}}_{t,\gamma} = e^{\bar{f}_t}\Big(\frac{d_{t,\gamma}}{d_{0,\gamma}}x_0 + \frac{e^{\bar{f}_T}\bar{g}_t^2}{d_{0,\gamma}}x_T + \big(\bar{h}_t - \frac{e^{2\bar{f}_T}\bar{h}_T\bar{g}_t^2}{d_{0,\gamma}}\big)\mathbf{m}\Big),
$$
$$
\bar{\sigma}_{s:t}^2 = e^{2\bar{f}_t}\bar{g}_{s:t}^2, \qquad \bar{\sigma}_t'^2 = \frac{\bar{\sigma}_t^2 \bar{\sigma}_{t:T}^2}{\bar{\sigma}_T^2}.
$$
$$
\tag{18}
$$

The derailed derivation is provided in Appendix A.4. Similar to (Yue et al., 2023) using the $l_1$ loss form to bring improved visual quality and details at the pixel level (Boyd, 2004; Hastie et al., 2017), we can derive the training objective. Denote $a_{t,\gamma} = e^{\bar{f}_t}d_{t,\gamma}$, assuming $\boldsymbol{\mu}_{t-1,\theta}$, $\sigma_{t-1,\theta}^2$ and $\boldsymbol{\mu}_{t-1,\gamma}$, $\sigma_{t-1,\gamma}^2$ are respectively the mean values and variances of $p_\theta(\mathbf{x}_{t-1} \mid \mathbf{x}_t, x_T)$ and $p(\mathbf{x}_{t-1} \mid \mathbf{x}_0, \mathbf{x}_t, x_T)$, suppose the score $\nabla_{\mathbf{x}_t} \log p(\mathbf{x}_t \mid x_T)$ is parameterized as $-\epsilon_\theta(\mathbf{x}_t, x_T, t)/\bar{\sigma}_t'$, the final training objective is as follows,

$$
\mathcal{L}_\theta = \mathbb{E}_{t,\mathbf{x}_0,\mathbf{x}_t,\mathbf{x}_T}\left[\frac{1}{2\sigma_{t-1,\theta}^2}\big\|\boldsymbol{\mu}_{t-1,\theta} - \boldsymbol{\mu}_{t-1,\gamma}\big\|_1\right],
$$
$$
\boldsymbol{\mu}_{t-1,\theta} = \mathbf{x}_t - f_t\mathbf{x}_t - h_t\mathbf{m} - g_t\mathbf{u}_{t,\gamma}^* + \frac{g_t^2}{\bar{\sigma}_t'}\epsilon_\theta(\mathbf{x}_t, x_T, t),
$$
$$
\boldsymbol{\mu}_{t-1,\gamma} = \bar{\boldsymbol{\mu}}_{t-1,\gamma} + \frac{\bar{\sigma}_{t-1}'^2 a_{t,\gamma}}{\bar{\sigma}_t'^2 a_{t-1,\gamma}}(\mathbf{x}_t - \bar{\boldsymbol{\mu}}_{t,\gamma}), \; \sigma_{t-1,\theta} = g_t.
$$
$$
\tag{19}
$$

Please refer to Appendix A.5 for detailed derivations. Therefore, we can recover or generate the origin image $\hat{x}_0$ through Euler sampling iterations. So far, we've built the UniDB framework, which establishes and expands the forward and

backward process of the diffusion bridge model through SOC and comprises Doob's $h$-transform as a special case.

## 4.4. UniDB unifies diffusion bridge models

Our UniDB is a unified framework for existing diffusion bridge models: DDBMs (VE) (Zhou et al., 2023), DDBMs (VP) (Zhou et al., 2023) and GOUB (Yue et al., 2023).

**Proposition 4.4.** *UniDB encompasses existing diffusion bridge models by employing different hyper-parameter spaces $\mathcal{H}$ as follows:*

- *DDBMs (VE) corresponds to UniDB with hyper-parameter $\mathcal{H}_{VE}(f_t = 0, h_t = 0, \gamma \to \infty)$*

- *DDBMs (VP) corresponds to UniDB with hyper-parameter $\mathcal{H}_{VP}(f_t = -\frac{1}{2}g_t^2, h_t = 0, \gamma \to \infty)$*

- *GOUB corresponds to UniDB with hyper-parameter $\mathcal{H}_{GOU}(f_t = \theta_t, h_t = -\theta_t, \mathbf{m} = \boldsymbol{\mu}, \gamma \to \infty)$*

Details of the proposition 4.4 are provided in Appendix A.6.

## 4.5. An Example: UniDB-GOU

It is evident that these diffusion bridge models like DDBMs (VE), DDBMs (VP) and GOUB all based on Doob's $h$-transform are all special cases of UniDB with $\gamma \to \infty$. However, according to Proposition 4.3, these models are not the effective choices. Therefore, we introduce UniDB based on the GOU process (5), hereafter referred to as UniDB-GOU, which retains the penalty coefficient $\gamma$ as the hyper-parameter. Considering the SOC problem with GOU process (5), the optimally controlled forward SDE is:

$$d\mathbf{x}_t = \left(\theta_t + g_t^2 \frac{e^{-2\bar{\theta}_{t:T}}}{\gamma^{-1} + \bar{\sigma}_{t:T}^2}\right)(x_T - \mathbf{x}_t)dt + g_t d\mathbf{w}_t, \quad (20)$$

and the mean value of forward transition $p(\mathbf{x}_t \mid x_0, x_T)$ is

$$\bar{\boldsymbol{\mu}}_{t,\gamma} = e^{-\bar{\theta}_t}\frac{1 + \gamma\bar{\sigma}_{t:T}^2}{1 + \gamma\bar{\sigma}_T^2}x_0 + \left(1 - e^{-\bar{\theta}_t}\frac{1 + \gamma\bar{\sigma}_{t:T}^2}{1 + \gamma\bar{\sigma}_T^2}\right)x_T. \quad (21)$$

Please refer to Appendix A.7 for detailed proof.

**Remark 1.** It's worth noting that our UniDB model can be a plugin module to the existing diffusion bridge with Doob's $h$-transform. Taking UniDB-GOU as an example, we highlight the key difference between UniDB-GOU and GOUB (the coefficient of $x_0$ in the mean value of forward transition and $h$-function term) as follows:

$$e^{-\bar{\theta}_t}\frac{\bar{\sigma}_{t:T}^2}{\bar{\sigma}_T^2} \Rightarrow e^{-\bar{\theta}_t}\frac{\gamma^{-1} + \bar{\sigma}_{t:T}^2}{\gamma^{-1} + \bar{\sigma}_T^2}$$

$$\underbrace{g_t\mathbf{h} = \frac{g_t e^{-2\bar{\theta}_{t:T}}(x_T - \mathbf{x}_t)}{\bar{\sigma}_{t:T}^2}}_{\text{GOUB}} \Rightarrow \underbrace{\mathbf{u}_{t,\gamma}^* = \frac{g_t e^{-2\bar{\theta}_{t:T}}(x_T - \mathbf{x}_t)}{\gamma^{-1} + \bar{\sigma}_{t:T}^2}}_{\text{UniDB-GOU}}$$

$$(22)$$

---

**Algorithm 1** UniDB Training

**repeat**

    Take a pair of images $\mathbf{x}_0 = x_0$ and $\mathbf{x}_T = x_T$

    $t \sim \text{Uniform}(\{1, ..., T\})$

    $\sigma_{t-1,\theta} = g_t$

    $a_{t,\gamma} = e^{-\bar{\theta}_t}\frac{\bar{\sigma}_{t:T}^2}{\bar{\sigma}_T^2} \quad \leftarrow \quad \text{GOUB}$

    $a_{t,\gamma} = e^{-\bar{\theta}_t}\frac{\gamma^{-1} + \bar{\sigma}_{t:T}^2}{\gamma^{-1} + \bar{\sigma}_T^2} \quad \leftarrow \quad \text{UniDB-GOU}$

    $\mathbf{x}_t = a_{t,\gamma}x_0 + (1 - a_{t,\gamma})x_T + \bar{\sigma}_t'\epsilon$

    $\bar{\boldsymbol{\mu}}_{t,\gamma} = a_{t,\gamma}x_0 + (1 - a_{t,\gamma})x_T$

    $\boldsymbol{\mu}_{t-1,\theta} = \mathbf{x}_t - \left(\theta_t + g_t^2\frac{e^{-2\bar{\theta}_{t:T}}}{\bar{\sigma}_{t:T}^2}\right)(x_T - \mathbf{x}_t)$

    $\qquad + \frac{g_t^2}{\bar{\sigma}_t'^2}\boldsymbol{\epsilon}_\theta(\mathbf{x}_t, x_T, t) \leftarrow \text{GOUB}$

    $\boldsymbol{\mu}_{t-1,\theta} = \mathbf{x}_t - \left(\theta_t + g_t^2\frac{e^{-2\bar{\theta}_{t:T}}}{\gamma^{-1} + \bar{\sigma}_{t:T}^2}\right)(x_T - \mathbf{x}_t)$

    $\qquad + \frac{g_t^2}{\bar{\sigma}_t'^2}\boldsymbol{\epsilon}_\theta(\mathbf{x}_t, x_T, t) \leftarrow \text{UniDB-GOU}$

    $\boldsymbol{\mu}_{t-1,\gamma} = \bar{\boldsymbol{\mu}}_{t-1,\gamma} + \frac{\bar{\sigma}_{t-1}' a_{t,\gamma}}{\bar{\sigma}_t'^2 a_{t-1,\gamma}}(\mathbf{x}_t - \bar{\boldsymbol{\mu}}_{t,\gamma})$

    Take gradient descent step on $\nabla_\theta \mathcal{L}_\theta$

**until** converged

---

Hence, only a few lines of code need to be adjusted to generate more realistic images using the same training method. We provide pseudo-code Algorithm 1 and Algorithm 2 for the training and sampling process of UniDB-GOU, respectively. The two algorithms encapsulate the core methodologies employed by our model to learn and explain how to restore HQ images from LQ images. Also, the red and the green parts highlight the main difference between UniDB and GOUB. Beyond the GOUB model, our UniDB framework can be similarly extended to other diffusion bridge models, such as DDBMs (VE) and DDBMs (VP). For detailed information on UniDB-VE and UniDB-VP, please refer to Appendix A.8.

---

**Algorithm 2** UniDB Sampling

**Input:** Low-Quality images $\mathbf{x}_T = x_T$.

**for** $t = T$ to $1$ **do**

    $z \sim N(0, I)$ if $t > 1$, else $z = 0$

    $\mathbf{x}_{t-1} = \mathbf{x}_t - \left(\theta_t + g_t^2\frac{e^{-2\bar{\theta}_{t:T}}}{\bar{\sigma}_{t:T}^2}\right)(x_T - \mathbf{x}_t)$

    $\qquad + \frac{g_t^2}{\bar{\sigma}_t'^2}\boldsymbol{\epsilon}_\theta(\mathbf{x}_t, x_T, t) - g_t z \leftarrow \text{GOUB}$

    $\mathbf{x}_{t-1} = \mathbf{x}_t - \left(\theta_t + g_t^2\frac{e^{-2\bar{\theta}_{t:T}}}{\gamma^{-1} + \bar{\sigma}_{t:T}^2}\right)(x_T - \mathbf{x}_t)$

    $\qquad + \frac{g_t^2}{\bar{\sigma}_t'^2}\boldsymbol{\epsilon}_\theta(\mathbf{x}_t, x_T, t) - g_t z \leftarrow \text{UniDB-GOU}$

**end for**

**Return** High-Quality images $\tilde{\mathbf{x}}_0$

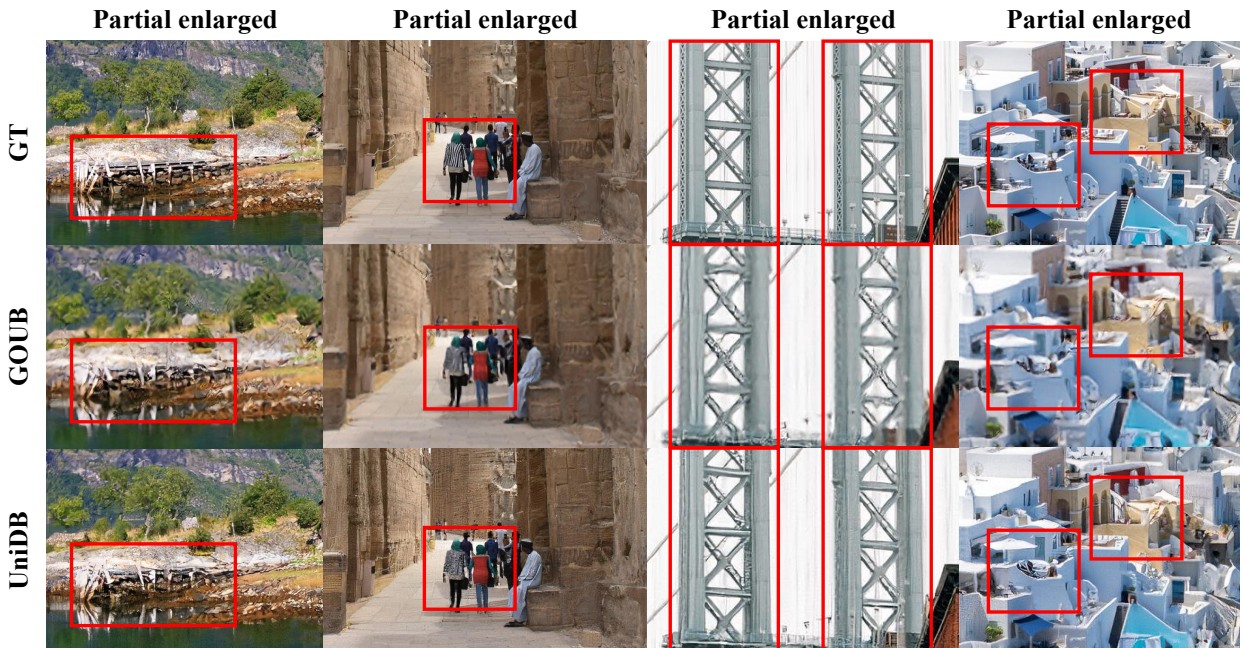

Figure 2. Qualitative comparison of visual results between GOUB (SDE) and UniDB (SDE) on DIV2K with zoomed-in image local regions (UniDB based on GOU process).

Building upon equations (20) and (21), we further present a proposition to characterize how the penalty coefficient $\gamma$ affects the controlled terminal distribution as follows:

**Proposition 4.5.** *Denote the initial state distribution $x_0$, the terminal distribution $\mathbf{x}_T^u$ by the controller and the pre-defined terminal distribution $x_T$, then*

$$\|\mathbf{x}_T^u - x_T\|_2^2 = \frac{e^{-2\bar{\theta}_T}}{\left(1 + \gamma\lambda^2(1 - e^{-2\bar{\theta}_T})\right)^2}\|x_T - x_0\|_2^2. \quad (23)$$

The detailed derivations of proposition 4.5 are provided in Appendix A.9. Notably, as $\gamma$ approaches infinity, the control terminal converges to the predefined endpoint. However, as analyzed in Proposition 4.3, this can result in suboptimal outcomes with blurry or overly smoothed image details. To address this, it is crucial to balance the control cost and terminal term by selecting the value of $\gamma$. In the following section, we will present comprehensive experiments to evaluate the impact of different $\gamma$ values on the results.

## 5. Experiments

In this section, we evaluate our models in image restoration tasks including Image 4×Super-resolution, Image De-raining, and Image Inpainting. We take four evaluation metrics: Peak Signal-to-Noise Ratio (PSNR, higher is better) (Fardo et al., 2016), Structural Similarity Index (SSIM, higher is better) (Wang et al., 2004), Learned Perceptual Image Patch Similarity (LPIPS, lower is better) (Zhang et al.,

2018) and Fréchet Inception Distance (FID, lower is better) (Heusel et al., 2017). For simple expressions in the following sections, UniDB (SDE) and UniDB (ODE) are applied to represent the UniDB-GOU with reverse SDE and reverse Mean-ODE, respectively. Please refer to Appendix B and C for all related implementation details and more experiment results, respectively.

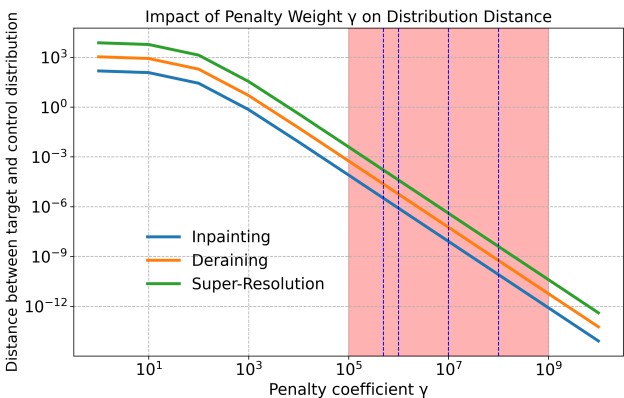

Figure 4. The distances between target and controlled terminal distributions for different datasets (CelebA-HQ, Rain100H, and DIV2K) with different penalty coefficients $\gamma$. The red shaded area and blue dotted line highlight our choice of $\gamma$.

### 5.1. Experiments Setup

According to Proposition 4.5, we first quantitatively analyze the $l_2$-norm distances between the two terminal distributions depicted in Figure 4. We computed the average distances

*Table 1.* Qualitative comparison with the relevant baselines on DIV2K, Rain100H, and CelebA-HQ 256×256 datasets.

| METHOD | Image Super-Resolution | | | | METHOD | Image Deraining | | | | METHOD | Image Inpainting | | | |
|---|---|---|---|---|---|---|---|---|---|---|---|---|---|---|
| | PSNR↑ | SSIM↑ | LPIPS↓ | FID↓ | | PSNR↑ | SSIM↑ | LPIPS↓ | FID↓ | | PSNR↑ | SSIM↑ | LPIPS↓ | FID↓ |
| Bicubic | 26.70 | 0.774 | 0.425 | 36.18 | MAXIM | 30.81 | 0.902 | 0.133 | 58.72 | PromptIR | 30.22 | 0.918 | 0.068 | 32.69 |
| DDRM | 24.35 | 0.592 | 0.364 | 78.71 | MHNet | 31.08 | 0.899 | 0.126 | 57.93 | DDRM | 27.16 | 0.899 | 0.089 | 37.02 |
| IR-SDE | 25.90 | 0.657 | 0.231 | 45.36 | IR-SDE | 31.65 | 0.904 | 0.047 | 18.64 | IR-SDE | 28.37 | 0.916 | 0.046 | 25.13 |
| GOUB (SDE) | 26.89 | 0.7478 | 0.220 | 20.85 | GOUB (SDE) | 31.96 | 0.9028 | 0.046 | 18.14 | GOUB (SDE) | 28.98 | 0.9067 | 0.037 | 4.30 |
| GOUB (ODE) | 28.50 | 0.8070 | 0.328 | 22.14 | GOUB (ODE) | 34.56 | 0.9414 | 0.077 | 32.83 | GOUB (ODE) | 31.39 | 0.9392 | 0.052 | 12.24 |
| UniDB (SDE) | 25.46 | 0.6856 | **0.179** | **16.21** | UniDB (SDE) | 32.05 | 0.9036 | **0.045** | 17.65 | UniDB (SDE) | 29.20 | 0.9077 | **0.036** | **4.08** |
| UniDB (ODE) | **28.64** | **0.8072** | 0.323 | 22.32 | UniDB (ODE) | **34.68** | **0.9426** | 0.074 | 31.16 | UniDB (ODE) | **31.67** | **0.9395** | 0.052 | 11.98 |

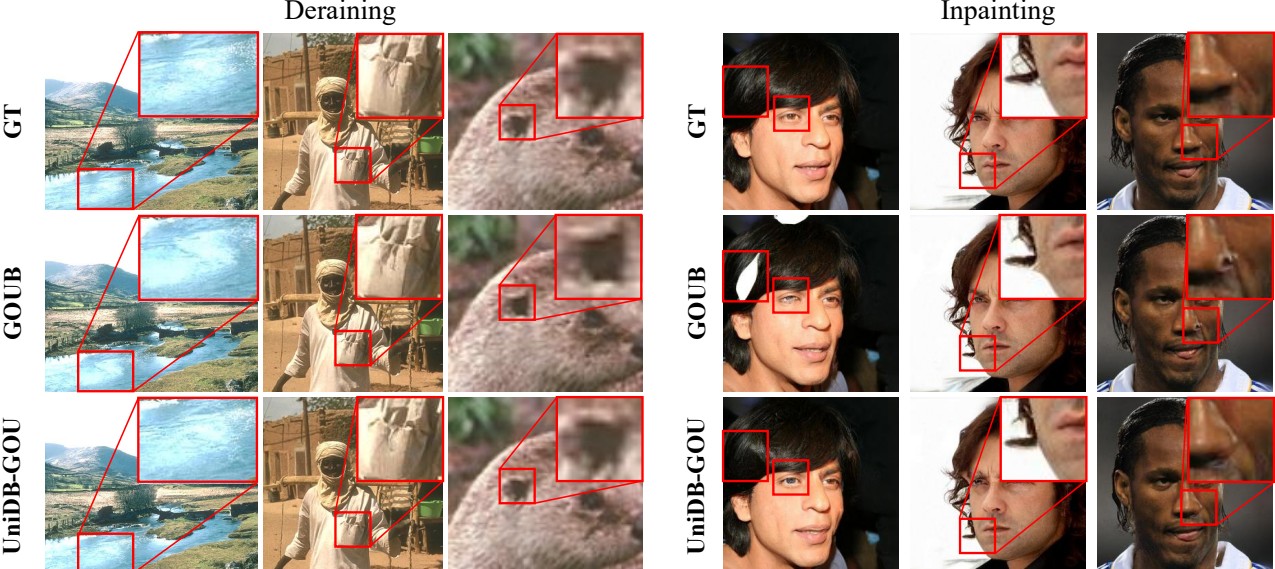

*Figure 3.* Qualitative comparison of visual results between GOUB (SDE) and UniDB (SDE) on the Rain100H dataset on Image Deraining (Left) and CelebA-HQ dataset on Image Inpainting (Right) with zoomed-in image local regions (UniDB based on GOU process).

between high-quality and low-quality images in the three datasets (CelebA-HQ, Rain100H, and DIV2K) related to the subsequent experimental section as the distances $\|x_T - x_0\|_2^2$ in (23). As can be seen, for all three datasets, these distances remain relatively small, ranging from $10^{-4}$ to $10^{-10}$ when $\gamma$ is within the range of $1 \times 10^5$ to $1 \times 10^9$. Therefore, our subsequent experiments will focus on the $\gamma$ of this range to further investigate the performance of UniDB-GOU.

### 5.2. Experimental Details

**Image 4×Super-Resolution Tasks.** In super-resolution, we evaluated our models based on DIV2K dataset (Agustsson & Timofte, 2017), which contains 2K-resolution high-quality images. During the experiment, all low-resolution images were 4× bicubic upscaling to the same image size as the paired high-resolution images. For comparison, we choose Bicubic interpolation (Kawar et al., 2022), DDRM (Kawar et al., 2022), IR-SDE (Luo et al., 2023), GOUB (SDE) (Yue et al., 2023) and GOUB (Mean-ODE) (Yue et al., 2023) following abbreviated as GOUB (ODE) as the baselines. The qualitative and quantitative results are illustrated in Table 1

and Figure 2. Visually, our proposed model demonstrates a significant improvement over the baseline across various metrics. It also excels by delivering superior performance in both visual quality and detail compared to other results.

**Image Deraining Tasks.** For image deraining tasks, we conducted the experiments based on Rain100H datasets (Yang et al., 2017). Particularly, to be consistent with other deraining models (Ren et al., 2019; Zamir et al., 2021; Luo et al., 2023; Yue et al., 2023), PSNR and SSIM scores on the Y channel (YCbCr space) are selected instead of the origin PSNR and SSIM. MAXIM (Tu et al., 2022), MHNet (Gao et al., 2025), IR-SDE (Luo et al., 2023), GOUB (SDE) (Yue et al., 2023) and GOUB (ODE) (Yue et al., 2023) are chosen as the baselines. The relevant experimental results are shown in the Table 1 and Figure 3. Similarly, our model achieved state-of-the-art results in the deraining task. Visually, it can also be observed that our model excels in capturing details such as the eyebrows, eye bags, and lips.

**Image Inpainting Tasks.** In image inpainting tasks, we evaluated our methods on CelebA-HQ 256×256 datasets

*Table 2.* Quantitative evaluation results for DIV2K, CelebA-HQ and Rain100H of UniDB-GOU with different penalty coefficients $\gamma$.

| TASKS | METRICS | Different $\gamma$ | | | | |
|---|---|---|---|---|---|---|
| | | $5 \times 10^5$ | $1 \times 10^6$ | $1 \times 10^7$ | $1 \times 10^8$ | $\infty$ |
| **Image 4×Super-Resolution** | **PSNR↑** | 24.94 | 24.72 | 25.46 | 25.06 | **26.89** |
| | **SSIM↑** | 0.6419 | 0.6587 | 0.6856 | 0.6393 | **0.7478** |
| | **LPIPS↓** | 0.234 | 0.199 | **0.179** | 0.289 | 0.220 |
| | **FID↓** | 20.33 | 18.37 | **16.21** | 23.76 | 20.85 |
| **Image Inpainting** | **PSNR↑** | 28.73 | 29.15 | **29.20** | 28.65 | 28.98 |
| | **SSIM↑** | 0.9065 | 0.9068 | **0.9077** | 0.9062 | 0.9067 |
| | **LPIPS↓** | 0.038 | 0.036 | **0.036** | 0.039 | 0.037 |
| | **FID↓** | 4.49 | 4.12 | **4.08** | 4.64 | 4.30 |
| **Image Deraining** | **PSNR↑** | 29.44 | 31.96 | 32.00 | **32.05** | 31.96 |
| | **SSIM↑** | 0.8715 | 0.9018 | 0.9029 | **0.9036** | 0.9028 |
| | **LPIPS↓** | 0.058 | 0.045 | 0.046 | **0.045** | 0.046 |
| | **FID↓** | 24.96 | 18.37 | 17.87 | **17.65** | 18.14 |

(Karras, 2017). For comparison, we choose DDRM (Kawar et al., 2022), PromptIR (Potlapalli et al., 2023), IR-SDE (Luo et al., 2023), GOUB (SDE) (Yue et al., 2023) and GOUB (ODE) (Yue et al., 2023) as the baselines. As for mask type, we take 100 thin masks consistent with the baselines. The relevant experimental results are shown in Table 1 and Figure 3. It is observed that our model achieved state-of-the-art results in all indicators and also delivered highly competitive outcomes on other metrics. From a visual perspective, our model excels in capturing details such as faces, eyes, chins, and noses.

### 5.3. Ablation Study

**Penalty Coefficient $\gamma$.** To evaluate the specific impact of different penalty coefficients $\gamma$ on model performance, we conducted the experiments above with several different $\gamma$. The final results are shown in Table 2. The results across all tasks show that the choice of $\gamma$ significantly influences the model's performance on all tasks, different optimal $\gamma$ for different tasks, and our UniDB achieves the best performance in almost all metrics. Particularly in super-resolution tasks, we focus more on the significantly better perceptual scores (LPIPS and FID) (Luo et al., 2023), demonstrating that UniDB ensures to capture and preserve more intricate image details and features as shown in Figure 2. These findings underscore the importance of carefully tuning $\gamma$ to achieve the best performance for specific tasks.

## 6. Conclusion

In this paper, we presented UniDB, a unified diffusion bridge framework based on stochastic optimal control principles, offering a novel perspective on diffusion bridges. Through this framework, we unify and extend existing diffu-

sion bridge models with Doob's $h$-transform like DDBMs and GOUB. Moreover, we demonstrate that the diffusion bridge with Doob's $h$-transform can be viewed as a specific case within UniDB when the terminal penalty coefficient approaches infinity. This insight helps elucidate why Doob's $h$-transform may lead to suboptimal image restoration, often resulting in blurred or distorted details. By simply adjusting this terminal penalty coefficient, UniDB achieves a marked improvement in image quality with minimal code modifications. Our experimental results underscore UniDB's superiority and versatility across various image processing tasks, particularly in enhancing image details for more realistic outputs. Despite these advantages, UniDB, like other standard diffusion bridge models, faces the challenge of computationally intensive sampling processes, especially with high-resolution images or complex restoration tasks. Future work will focus on developing strategies to accelerate the sampling process, enhancing UniDB's practicality, particularly for real-time applications.

## Acknowledgement

This work was supported by National Natural Science Foundation of China (62303319, 62406195), Shanghai Local College Capacity Building Program (23010503100), ShanghaiTech AI4S Initiative SHTAI4S202404, HPC Platform of ShanghaiTech University, Core Facility Platform of Computer Science and Communication of ShanghaiTech University, and MoE Key Laboratory of Intelligent Perception and Human-Machine Collaboration (ShanghaiTech University), and Shanghai Engineering Research Center of Intelligent Vision and Imaging.

## Impact Statement

There are many potential societal consequences of our work, none of which we feel must be specifically highlighted here.

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

# Appendix Contents

# A. Proof

## A.1. Proof of Theorem 4.1

**Theorem 4.1.** *Consider the SOC problem* (12), *denote* $d_{t,\gamma} = \gamma^{-1} + e^{2\bar{f}_T} \bar{g}_{t:T}^2$, $\bar{f}_{s:t} = \int_s^t f_z dz$, $\bar{h}_{s:t} = \int_s^t e^{-\bar{f}_z} h_z dz$ *and* $\bar{g}_{s:t}^2 = \int_s^t e^{-2\bar{f}_z} g_z^2 dz$, *denote* $\bar{f}_t$, $\bar{h}_t$ *and* $\bar{g}_t^2$ *for simplification when* $s = 0$, *then the closed-form optimal controller* $\mathbf{u}_{t,\gamma}^*$ *is*

$$\mathbf{u}_{t,\gamma}^* = g_t e^{\bar{f}_{t:T}} \frac{x_T - e^{\bar{f}_{t:T}} \mathbf{x}_t - \mathbf{m} e^{\bar{f}_T} \bar{h}_{t:T}}{d_{t,\gamma}}, \tag{13}$$

*and the transition of* $\mathbf{x}_t$ *from* $x_0$ *and* $x_T$ *is*

$$\mathbf{x}_t = e^{\bar{f}_t} \left( \frac{d_{t,\gamma}}{d_{0,\gamma}} x_0 + \frac{e^{\bar{f}_T} \bar{g}_t^2}{d_{0,\gamma}} x_T + \left( \bar{h}_t - \frac{e^{2\bar{f}_T} \bar{h}_T \bar{g}_t^2}{d_{0,\gamma}} \right) \mathbf{m} \right). \tag{14}$$

*Proof.* According to Pontryagin Maximum Principle (Levine, 1972; Kirk, 2004) recipe, one can construct the Hamiltonian:

$$H(t, \mathbf{x}_t, \mathbf{u}_{t,\gamma}, \mathbf{p}_t) = \frac{1}{2} \|\mathbf{u}_{t,\gamma}\|_2^2 + \mathbf{p}_t^T \left( f_t \mathbf{x}_t + h_t \mathbf{m} + g_t \mathbf{u}_t \right). \tag{24}$$

By setting:

$$\frac{\partial H}{\partial \mathbf{u}_{t,\gamma}} = 0 \quad \Rightarrow \quad \mathbf{u}_{t,\gamma}^* = -g_t \mathbf{p}_t. \tag{25}$$

Then the value function becomes

$$V^* = \min_{\mathbf{u}_{t,\gamma}} H(t, \mathbf{x}_t, \mathbf{p}_t, \mathbf{u}_{t,\gamma}) = H(t, \mathbf{x}_t, \mathbf{p}_t, \mathbf{u}_{t,\gamma}^*) = -\frac{g_t^2}{2} \|\mathbf{p}_t\|_2^2 + f_t \mathbf{p}_t^T \mathbf{x}_t + h_t \mathbf{p}_t^T \mathbf{m}. \tag{26}$$

Now, according to minimum principle theorem to obtain the following set of differential equations:

$$\frac{d\mathbf{x}_t}{dt} = \nabla_{\mathbf{p}_t} H \left( \mathbf{x}_t, \mathbf{p}_t, \mathbf{u}_{t,\gamma}^*, t \right) = -g_t^2 \mathbf{p}_t + f_t \mathbf{x}_t + h_t \mathbf{m}, \tag{27}$$

$$\frac{d\mathbf{p}_t}{dt} = -\nabla_{\mathbf{x}_t} H \left( \mathbf{x}_t, \mathbf{p}_t, \mathbf{u}_t^*, t \right) = -\mathbf{p}_t f_t, \tag{28}$$

$$\mathbf{x}_0 = x_0, \tag{29}$$

$$\mathbf{p}_T = \gamma \left( \mathbf{x}_T - x_T \right). \tag{30}$$

Solving the Equation (28), we have:

$$\mathbf{p}_t = \mathbf{p}_0 e^{-\bar{f}_t},$$
$$\mathbf{p}_T = \mathbf{p}_0 e^{-\bar{f}_T}. \tag{31}$$

Solve the Equation (27):

$$\frac{\mathrm{d}\mathbf{x}_t}{\mathrm{d}t} = f_t \mathbf{x}_t + h_t \mathbf{m} - g_t^2 \mathbf{p}_t$$
$$\Rightarrow \quad \frac{\mathrm{d}(e^{-\bar{f}_t}\mathbf{x}_t)}{\mathrm{d}t} = e^{-\bar{f}_t} h_t \mathbf{m} - e^{-\bar{f}_t} g_t^2 \mathbf{p}_t,$$
$$\Rightarrow \quad e^{-\bar{f}_t}\mathbf{x}_t - \mathbf{x}_0 = \mathbf{m}\bar{h}_t - \mathbf{p}_0 \bar{g}_t^2,$$
$$\Rightarrow \quad e^{-\bar{f}_t}\mathbf{x}_t - x_0 = \mathbf{m}\bar{h}_t - \mathbf{p}_0 \bar{g}_t^2.$$

Hence, we can get:

$$\mathbf{x}_T = e^{\bar{f}_T} x_0 + \mathbf{m} e^{\bar{f}_T} \bar{h}_T - \mathbf{p}_T e^{2\bar{f}_T} \bar{g}_T^2, \tag{32}$$

and

$$\mathbf{x}_t = e^{\bar{f}_t} x_0 + \mathbf{m} e^{\bar{f}_t} \bar{h}_t - \mathbf{p}_T e^{\bar{f}_t} e^{\bar{f}_T} \bar{g}_t^2. \tag{33}$$

Take the Equation (32) into the Equation (30) and solve $\mathbf{p}_T$,

$$\mathbf{p}_T = \gamma \left( e^{\bar{f}_T} x_0 + \mathbf{m} e^{\bar{f}_T} \bar{h}_T - \mathbf{p}_T e^{2\bar{f}_T} \bar{g}_T^2 - x_T \right) \tag{34}$$

$$\Rightarrow \quad \mathbf{p}_T = \frac{\gamma \left( e^{\bar{f}_T} x_0 + \mathbf{m} e^{\bar{f}_T} \bar{h}_T - x_T \right)}{1 + \gamma e^{2\bar{f}_T} \bar{g}_T^2}. \tag{35}$$

Also, take the Equation (34) into the equation (33),

$$\mathbf{x}_t = e^{\bar{f}_t} x_0 + \mathbf{m} e^{\bar{f}_t} \bar{h}_t - e^{\bar{f}_t} e^{\bar{f}_T} \bar{g}_t^2 \frac{e^{\bar{f}_T} x_0 + \mathbf{m} e^{\bar{f}_T} \bar{h}_T - x_T}{\gamma^{-1} + e^{2\bar{f}_T} \bar{g}_T^2}$$
$$= e^{\bar{f}_t} \left( \frac{d_{t,\gamma}}{d_{0,\gamma}} x_0 + \frac{e^{\bar{f}_T} \bar{g}_t^2}{d_{0,\gamma}} x_T + \left( \bar{h}_t - \frac{e^{2\bar{f}_T} \bar{h}_T \bar{g}_t^2}{d_{0,\gamma}} \right) \mathbf{m} \right). \tag{36}$$

Preserve $\gamma$,

$$\mathbf{u}_{t,\gamma}^* = -g_t \mathbf{p}_t$$
$$= -g_t e^{-\bar{f}_t} e^{\bar{f}_T} \frac{e^{\bar{f}_T} x_0 + \mathbf{m} e^{\bar{f}_T} \bar{h}_T - x_T}{\gamma^{-1} + e^{2\bar{f}_T} \bar{g}_T^2}$$
$$= g_t e^{\bar{f}_{t:T}} \frac{x_T - e^{\bar{f}_{t:T}} \mathbf{x}_t - \mathbf{m} e^{\bar{f}_T} \bar{h}_{t:T}}{d_{t,\gamma}}, \tag{37}$$

with the fact (36)

$$\mathbf{x}_t = e^{\bar{f}_t} \left( \frac{d_{t,\gamma}}{d_{0,\gamma}} x_0 + \frac{e^{\bar{f}_T} \bar{g}_t^2}{d_{0,\gamma}} x_T + \left( \bar{h}_t - \frac{e^{2\bar{f}_T} \bar{h}_T \bar{g}_t^2}{d_{0,\gamma}} \right) \mathbf{m} \right), \tag{38}$$

which concludes the proof of the Proposition 4.1. □

## A.2. Proof of Theorem 4.2

**Theorem 4.2.** *For the SOC problem* (12)*, when $\gamma \to \infty$, the optimal controller becomes $\mathbf{u}_{t,\infty}^* = g_t \nabla_{\mathbf{x}_t} \log p(\mathbf{x}_T \mid \mathbf{x}_t)$, and the corresponding forward and backward SDE with the linear SDE form* (10) *are the same as Doob's h-transform as in* (3) *and* (4)*.*

*Proof.* Since in Proposition 4.1 we have solved the control problem and the optimal controller $\mathbf{u}_{t,\infty}^*$ is:

$$\mathbf{u}_{t,\infty}^* = \lim_{\gamma \to \infty} \mathbf{u}_{t,\gamma}^* = g_t e^{\bar{f}_{t:T}} \frac{\mathbf{x}_T - e^{\bar{f}_{t:T}} \mathbf{x}_t - \mathbf{m} e^{\bar{f}_T} \bar{h}_{t:T}}{e^{2\bar{f}_T} \bar{g}_{t:T}^2}. \tag{13}$$

Now we calculate the transition probability $p(\mathbf{x}_T \mid \mathbf{x}_t)$ and related $h$ function $\mathbf{h}(\mathbf{x}_t, t, \mathbf{x}_T, T)$.

Consider $F(\mathbf{x}_t, t) = \mathbf{x}_t e^{-\bar{f}_t}$, according to the Ito differential formula, we get:

$$\mathrm{d}F = -f_t \mathbf{x}_t e^{-\bar{f}_t} \mathrm{d}t + e^{-\bar{f}_t} \mathrm{d}\mathbf{x}_t \tag{39}$$

$$\Rightarrow \quad \mathrm{d}F = -f_t \mathbf{x}_t e^{-\bar{f}_t} \mathrm{d}t + e^{-\bar{f}_t} \Big( (f_t \mathbf{x}_t + h_t \mathbf{m}) \, \mathrm{d}t + g_t \mathrm{d}\mathbf{w}_t \Big), \tag{40}$$

$$\Rightarrow \quad \mathrm{d}F = h_t e^{-\bar{f}_t} \mathbf{m} \mathrm{d}t + e^{-\bar{f}_t} g_t \mathrm{d}\mathbf{w}_t, \tag{41}$$

$$\Rightarrow \quad \mathbf{x}_T e^{-\bar{f}_T} - \mathbf{x}_t e^{-\bar{f}_t} = \mathbf{m} \bar{h}_{t:T} + \int_t^T e^{-\bar{f}_z} g_z \mathrm{d}w_z, \tag{42}$$

$$\Rightarrow \quad \mathbf{x}_T \sim N \left( e^{\bar{f}_{t:T}} \mathbf{x}_t + \mathbf{m} e^{\bar{f}_T} \bar{h}_{t:T}, e^{2\bar{f}_T} \bar{g}_{t:T}^2 \mathbf{I} \right), \tag{43}$$

$$\Rightarrow \quad \nabla_{\mathbf{x}_t} \log p(\mathbf{x}_T \mid \mathbf{x}_t) = -\nabla_{\mathbf{x}_t} \frac{(\mathbf{x}_T - e^{\bar{f}_{t:T}} \mathbf{x}_t - \mathbf{m} e^{\bar{f}_T} \bar{h}_{t:T})^2}{2 e^{2\bar{f}_T} \bar{g}_{t:T}^2} = \frac{e^{\bar{f}_{t:T}} \left( \mathbf{x}_T - e^{\bar{f}_{t:T}} \mathbf{x}_t - \mathbf{m} e^{\bar{f}_T} \bar{h}_{t:T} \right)}{e^{2\bar{f}_T} \bar{g}_{t:T}^2}, \tag{44}$$

$$\Rightarrow \quad \mathbf{u}_{t,\infty}^* = g_t e^{\bar{f}_{t:T}} \frac{\mathbf{x}_T - e^{\bar{f}_{t:T}} \mathbf{x}_t - \mathbf{m} e^{\bar{f}_T} \bar{h}_{t:T}}{e^{2\bar{f}_T} \bar{g}_{t:T}^2} = g_t \nabla_{\mathbf{x}_t} \log p(\mathbf{x}_T \mid \mathbf{x}_t) = g_t \mathbf{h}(\mathbf{x}_t, t, \mathbf{x}_T, T). \tag{45}$$

The forward SDEs obtained through SOC and Doob's h-transform are both formed as

$$\mathrm{d}\mathbf{x}_t = \left( f_t \mathbf{x}_t + h_t \mathbf{m} + g_t^2 \frac{e^{\bar{f}_{t:T}} \left( \mathbf{x}_T - e^{\bar{f}_{t:T}} \mathbf{x}_t - \mathbf{m} e^{\bar{f}_T} \bar{h}_{t:T} \right)}{e^{2\bar{f}_T} \bar{g}_{t:T}^2} \right) \mathrm{d}t + g_t \mathrm{d}\mathbf{w}_t, \tag{46}$$

and the both backward SDEs are

$$\mathrm{d}\mathbf{x}_t = \left( f_t \mathbf{x}_t + h_t \mathbf{m} + g_t^2 \frac{e^{\bar{f}_{t:T}} \left( \mathbf{x}_T - e^{\bar{f}_{t:T}} \mathbf{x}_t - \mathbf{m} e^{\bar{f}_T} \bar{h}_{t:T} \right)}{e^{2\bar{f}_T} \bar{g}_{t:T}^2} - g_t^2 \nabla_{\mathbf{x}_t} p(\mathbf{x}_t \mid x_T) \right) \mathrm{d}t + g_t \mathrm{d}\mathbf{w}_t, \tag{47}$$

which concludes the proof of the Theorem 4.2.

$\square$

## A.3. Proof of Proposition 4.3

**Proposition 4.3** *Consider the SOC problem* (12)*, denote $\mathcal{J}(\mathbf{u}_{t,\gamma}, \gamma) \triangleq \int_0^T \frac{1}{2} \|\mathbf{u}_{t,\gamma}\|_2^2 \, \mathrm{d}t + \frac{\gamma}{2} \|\mathbf{x}_T^u - x_T\|_2^2$ as the overall cost of the system, $\mathbf{u}_{t,\gamma}^*$ as the optimal controller* (13)*, then*

$$\mathcal{J}(\mathbf{u}_{t,\gamma}^*, \gamma) \le \mathcal{J}(\mathbf{u}_{t,\infty}^*, \infty). \tag{48}$$

*Proof.* According to (13) and (14), denote $a = e^{\bar{f}_T} x_0 - x_T + \mathbf{m} e^{\bar{f}_T} \bar{h}_T$,

$$\mathbf{u}_{t,\gamma}^* = g_t e^{\bar{f}_{t:T}} \frac{x_T - e^{\bar{f}_{t:T}} \mathbf{x}_t - \mathbf{m} e^{\bar{f}_T} \bar{h}_{t:T}}{d_{t,\gamma}}$$

$$\Rightarrow \quad \mathbf{u}_{t,\gamma}^* = -g_t e^{-\bar{f}_t} e^{\bar{f}_T} \frac{e^{\bar{f}_T} x_0 + \mathbf{m} e^{\bar{f}_T} \bar{h}_T - x_T}{d_{t,\gamma}},$$

$$\Rightarrow \quad \|\mathbf{u}_{t,\gamma}^*\|_2^2 = g_t^2 e^{-2\bar{f}_t} e^{2\bar{f}_T} \frac{\|e^{\bar{f}_T} x_0 + \mathbf{m} e^{\bar{f}_T} \bar{h}_T - x_T\|_2^2}{d_{t,\gamma}^2},$$

$$\Rightarrow \quad \|\mathbf{u}_{t,\gamma}^*\|_2^2 = g_t^2 e^{-2\bar{f}_t} e^{2\bar{f}_T} \frac{\|a\|_2^2}{d_{t,\gamma}^2}.$$

Similarly,

$$\|\mathbf{u}_{t,\infty}^*\|_2^2 = g_t^2 e^{-2\bar{f}_t} e^{2\bar{f}_T} \frac{\|a\|_2^2}{(e^{2\bar{f}_T} \bar{g}_T^2)^2}. \tag{49}$$

Furthermore, take $t = T$ in (14),

$$\mathbf{x}_T^u = \left( \frac{\gamma^{-1} e^{\bar{f}_T}}{\gamma^{-1} + e^{2\bar{f}_T} \bar{g}_T^2} \right) x_0 + \left( \frac{e^{2\bar{f}_T} \bar{g}_T^2}{\gamma^{-1} + e^{2\bar{f}_T} \bar{g}_T^2} \right) x_T + e^{\bar{f}_T} \left( \frac{\gamma^{-1} \bar{h}_T}{\gamma^{-1} + e^{2\bar{f}_T} \bar{g}_T^2} \right) \mathbf{m}, \tag{50}$$

which implies

$$
\begin{aligned}
\|\mathbf{x}_T^u - x_T\|_2^2 &= \| \left( \frac{\gamma^{-1} e^{\bar{f}_T}}{\gamma^{-1} + e^{2\bar{f}_T} \bar{g}_T^2} \right) x_0 + \left( \frac{e^{2\bar{f}_T} \bar{g}_T^2}{\gamma^{-1} + e^{2\bar{f}_T} \bar{g}_T^2} \right) x_T + e^{\bar{f}_T} \left( \frac{\gamma^{-1} \bar{h}_T}{\gamma^{-1} + e^{2\bar{f}_T} \bar{g}_T^2} \right) \mathbf{m} - x_T \|_2^2 \\
&= \| \left( \frac{\gamma^{-1} e^{\bar{f}_T}}{\gamma^{-1} + e^{2\bar{f}_T} \bar{g}_T^2} \right) x_0 - \left( \frac{\gamma^{-1}}{\gamma^{-1} + e^{2\bar{f}_T} \bar{g}_T^2} \right) x_T + e^{\bar{f}_T} \left( \frac{\gamma^{-1} \bar{h}_T}{\gamma^{-1} + e^{2\bar{f}_T} \bar{g}_T^2} \right) \mathbf{m} \|_2^2 \\
&= \| \left( \frac{e^{\bar{f}_T}}{1 + \gamma e^{2\bar{f}_T} \bar{g}_T^2} \right) x_0 - \left( \frac{1}{1 + \gamma e^{2\bar{f}_T} \bar{g}_T^2} \right) x_T + e^{\bar{f}_T} \left( \frac{\bar{h}_T}{1 + \gamma e^{2\bar{f}_T} \bar{g}_T^2} \right) \mathbf{m} \|_2^2 \\
&= \frac{\|e^{\bar{f}_T} x_0 - x_T + \mathbf{m} e^{\bar{f}_T} \bar{h}_T\|_2^2}{(1 + \gamma e^{2\bar{f}_T} \bar{g}_T^2)^2} = \frac{\|a\|_2^2}{(1 + \gamma e^{2\bar{f}_T} \bar{g}_T^2)^2},
\end{aligned}
\tag{51}
$$

and

$$\lim_{\gamma \to \infty} \frac{\gamma}{2} \|\mathbf{x}_T^u - x_T\|_2^2 = \lim_{\gamma \to \infty} \frac{\gamma}{2(1 + \gamma e^{2\bar{f}_T} \bar{g}_T^2)^2} \|a\|_2^2 = 0. \tag{52}$$

Hence,

$$
\begin{aligned}
\frac{1}{2} \int_0^T \left( \|\mathbf{u}_{t,\infty}^*\|_2^2 - \|\mathbf{u}_{t,\gamma}^*\|_2^2 \right) dt &= \frac{1}{2} e^{2\bar{f}_T} \|a\|_2^2 \bar{g}_T^2 \left( \frac{1}{(e^{2\bar{f}_T} \bar{g}_T^2)^2} - \frac{1}{(\gamma^{-1} + e^{2\bar{f}_T} \bar{g}_T^2)^2} \right) \\
&= \frac{1}{2} e^{2\bar{f}_T} \|a\|_2^2 \bar{g}_T^2 \frac{1 + 2\gamma e^{2\bar{f}_T} \bar{g}_T^2}{(e^{2\bar{f}_T} \bar{g}_T^2)^2 (1 + \gamma e^{2\bar{f}_T} \bar{g}_T^2)^2} \\
&= \frac{1}{2} \frac{1 + 2\gamma e^{2\bar{f}_T} \bar{g}_T^2}{(e^{2\bar{f}_T} \bar{g}_T^2)(1 + \gamma e^{2\bar{f}_T} \bar{g}_T^2)^2} \|a\|_2^2 \\
&\geq \frac{1}{2} \frac{\gamma e^{2\bar{f}_T} \bar{g}_T^2}{(e^{2\bar{f}_T} \bar{g}_T^2)(1 + \gamma e^{2\bar{f}_T} \bar{g}_T^2)^2} \|a\|_2^2 \\
&= \frac{1}{2} \frac{\gamma}{(1 + \gamma e^{2\bar{f}_T} \bar{g}_T^2)^2} \|a\|_2^2 \\
&= \frac{\gamma}{2} \|\mathbf{x}_T^u - x_T\|_2^2 \\
&= \frac{\gamma}{2} \|\mathbf{x}_T^u - x_T\|_2^2 - \lim_{\gamma \to \infty} \frac{\gamma}{2} \|\mathbf{x}_T^u - x_T\|_2^2.
\end{aligned}
\tag{53}
$$

Therefore,

$$\frac{\gamma}{2}\|\mathbf{x}_T^u - x_T\|_2^2 - \lim_{\gamma \to \infty} \frac{\gamma}{2}\|\mathbf{x}_T^u - x_T\|_2^2 \leq \frac{1}{2}\int_0^T \left(\|\mathbf{u}_{t,\infty}^*\|_2^2 - \|\mathbf{u}_{t,\gamma}^*\|_2^2\right) dt \tag{54}$$

$$\Leftrightarrow \quad \frac{1}{2}\int_0^T \|\mathbf{u}_{t,\gamma}^*\|_2^2 dt + \frac{\gamma}{2}\|\mathbf{x}_T^u - x_T\|_2^2 \leq \frac{1}{2}\int_0^T \|\mathbf{u}_{t,\infty}^*\|_2^2 dt + \lim_{\gamma \to \infty} \frac{\gamma}{2}\|\mathbf{x}_T^u - x_T\|_2^2, \tag{55}$$

$$\Leftrightarrow \quad \mathcal{J}(\mathbf{u}_{t,\gamma}^*, \gamma) \leq \mathcal{J}(\mathbf{u}_{t,\infty}^*, \infty), \tag{56}$$

which concludes the proof of Proposition 4.3. $\qquad\square$

### A.4. Derivation of the transition probability (18)

*Suppose $\bar{\boldsymbol{\mu}}_{t,\gamma}$ and $\bar{\sigma}_t'$ denote the mean value and variance of the transition probability $p(\mathbf{x}_t \mid x_0, x_T)$, then*

$$p(\mathbf{x}_t \mid x_0, x_T) = \mathcal{N}(\bar{\boldsymbol{\mu}}_{t,\gamma}, \bar{\sigma}_t'^2 \mathbf{I}),$$

$$\bar{\boldsymbol{\mu}}_{t,\gamma} = e^{\bar{f}_t}\left(\frac{d_{t,\gamma}}{d_{0,\gamma}}x_0 + \frac{e^{\bar{f}_T}\bar{g}_t^2}{d_{0,\gamma}}x_T + \left(\bar{h}_t - \frac{e^{2\bar{f}_T}\bar{h}_T\bar{g}_t^2}{d_{0,\gamma}}\right)\mathbf{m}\right), \tag{18}$$

$$\bar{\sigma}_{s:t}^2 = e^{2\bar{f}_t}\bar{g}_{s:t}^2, \qquad \bar{\sigma}_t'^2 = \frac{\bar{\sigma}_t^2 \bar{\sigma}_{t:T}^2}{\bar{\sigma}_T^2}.$$

*Proof.* Since $\bar{\boldsymbol{\mu}}_{t,\gamma}$ remains the same as the closed-form relationship (14), we would focus on how to obtain $\bar{\sigma}_{s:t}^2$ and $\bar{\sigma}_t'^2$. In Equation (39) of Theorem 4.2, we've obtained:

$$p(\mathbf{x}_t \mid \mathbf{x}_s) \sim N\left(e^{\bar{f}_{s:t}}\mathbf{x}_s + \mathbf{m}e^{\bar{f}_t}\bar{h}_{s:t}, e^{2\bar{f}_t}\bar{g}_{s:t}^2\mathbf{I}\right),$$

$$\sim N\left(e^{\bar{f}_{s:t}}\mathbf{x}_s + \mathbf{m}e^{\bar{f}_t}\bar{h}_{s:t}, \bar{\sigma}_{s:t}^2\mathbf{I}\right). \tag{57}$$

Take $\bar{\sigma}_{s:t}^2 = e^{2\bar{f}_t}\bar{g}_{s:t}^2$ as the coefficient of the noise term, then, through Bayes' formula,

$$p(\mathbf{x}_t \mid x_0, x_T) = \frac{p(x_T \mid \mathbf{x}_t, x_0)p(\mathbf{x}_t \mid x_0)}{p(x_T \mid x_0)} = \frac{p(x_T \mid \mathbf{x}_t)p(\mathbf{x}_t \mid x_0)}{p(x_T \mid x_0)}$$

$$\Rightarrow \quad \bar{\sigma}_t'^2 = \frac{\bar{\sigma}_t^2 \bar{\sigma}_{t:T}^2}{\bar{\sigma}_T^2},$$

which concludes the derivation of the the transition probability (18). $\qquad\square$

### A.5. Derivation of the training objective (19)

*Denote $a_{t,\gamma} = e^{\bar{f}_t}d_{t,\gamma}$, assuming $\boldsymbol{\mu}_{t-1,\theta}, \sigma_{t-1,\theta}^2$ and $\boldsymbol{\mu}_{t-1,\gamma}, \sigma_{t-1,\gamma}^2$ are respectively the mean values and variances of $p_\theta(\mathbf{x}_{t-1} \mid \mathbf{x}_t, x_T)$ and $p(\mathbf{x}_{t-1} \mid x_0, \mathbf{x}_t, x_T)$, suppose the score $\nabla_{\mathbf{x}_t} \log p(\mathbf{x}_t \mid x_T)$ is parameterized as $-\boldsymbol{\epsilon}_\theta(\mathbf{x}_t, x_T, t)/\bar{\sigma}_t'$, the final training objective is as follows,*

$$\mathcal{L}_\theta = \mathbb{E}_{t,\mathbf{x}_0,\mathbf{x}_t,x_T}\left[\frac{1}{2\sigma_{t-1,\theta}^2}\|\boldsymbol{\mu}_{t-1,\theta} - \boldsymbol{\mu}_{t-1,\gamma}\|_1\right],$$

$$\boldsymbol{\mu}_{t-1,\theta} = \mathbf{x}_t - f_t\mathbf{x}_t - h_t\mathbf{m} - g_t\mathbf{u}_{t,\gamma}^* + \frac{g_t^2}{\bar{\sigma}_t'}\boldsymbol{\epsilon}_\theta(\mathbf{x}_t, x_T, t), \tag{19}$$

$$\boldsymbol{\mu}_{t-1,\gamma} = \bar{\boldsymbol{\mu}}_{t-1,\gamma} + \frac{\bar{\sigma}_{t-1}'^2 a_{t,\gamma}}{\bar{\sigma}_t'^2 a_{t-1,\gamma}}(\mathbf{x}_t - \bar{\boldsymbol{\mu}}_{t,\gamma}), \ \sigma_{t-1,\theta} = g_t.$$

*Proof.* Firstly, as for the training objective (19), according to GOUB (Yue et al., 2023):

$$
\begin{aligned}
\mathbb{E}_{p(\mathbf{x}_0)}[\log p_\theta(\mathbf{x}_0 \mid x_T)] \geq \mathbb{E}_{p(\mathbf{x}_0)}\Bigg[ & \mathbb{E}_{p(\mathbf{x}_1|\mathbf{x}_0)}\left[\log p_\theta\left(\mathbf{x}_0 \mid \mathbf{x}_1, x_T\right)\right] \\
& - \sum_{t=2}^{T} \mathbb{E}_{p(\mathbf{x}_t|\mathbf{x}_0)}[KL\left(p\left(\mathbf{x}_{t-1} \mid \mathbf{x}_0, \mathbf{x}_t, x_T\right) || p_\theta\left(\mathbf{x}_{t-1} \mid \mathbf{x}_t, x_T\right)\right)]\Bigg] \\
& = ELBO.
\end{aligned}
\tag{58}
$$

Accordingly,

$$
\begin{aligned}
& KL\left(p\left(\mathbf{x}_{t-1} \mid \mathbf{x}_0, \mathbf{x}_t, x_T\right) || p_\theta\left(\mathbf{x}_{t-1} \mid \mathbf{x}_t, x_T\right)\right) \\
& = \mathbb{E}_{p(\mathbf{x}_{t-1}|\mathbf{x}_0,\mathbf{x}_t,x_T)}\left[\log \frac{\frac{1}{\sqrt{2\pi}\sigma_{t-1}}e^{-(\mathbf{x}_{t-1}-\boldsymbol{\mu}_{t-1,\gamma})^2/2\sigma_{t-1}^2}}{\frac{1}{\sqrt{2\pi}\sigma_{\theta,t-1}}e^{-(\mathbf{x}_{t-1}-\boldsymbol{\mu}_{\theta,t-1})^2/2\sigma_{\theta,t-1}^2}}\right] \\
& = \mathbb{E}_{p(\mathbf{x}_{t-1}|\mathbf{x}_0,\mathbf{x}_t,x_T)}\left[\log \sigma_{\theta,t-1} - \log \sigma_{t-1} - (\mathbf{x}_{t-1} - \boldsymbol{\mu}_{t-1,\gamma})^2/2\sigma_{t-1}^2 + (\mathbf{x}_{t-1} - \boldsymbol{\mu}_{\theta,t-1})^2/2\sigma_{\theta,t-1}^2\right] \\
& = \log \sigma_{\theta,t-1} - \log \sigma_{t-1} - \frac{1}{2} + \frac{\sigma_{t-1}^2}{2\sigma_{\theta,t-1}^2} + \frac{(\boldsymbol{\mu}_{t-1,\gamma} - \boldsymbol{\mu}_{\theta,t-1})^2}{2\sigma_{\theta,t-1}^2}.
\end{aligned}
\tag{59}
$$

Hence, we ignore some constants and minimizing the negative ELBO, leading to the training objective:

$$
\mathcal{L} = \mathbb{E}_{t,\mathbf{x}_0,\mathbf{x}_t,x_T}\left[\frac{1}{2\sigma_{t-1,\theta}^2}\|\boldsymbol{\mu}_{t-1,\theta} - \boldsymbol{\mu}_{t-1,\gamma}\|\right],
\tag{60}
$$

Then, as for solving the closed form of $\boldsymbol{\mu}_{t-1,\theta}$, $\sigma_{t-1,\theta}^2$ and $\boldsymbol{\mu}_{t-1,\gamma}$, through Bayes' formula,

$$
\begin{aligned}
p\left(\mathbf{x}_{t-1} \mid x_0, \mathbf{x}_t, x_T\right) & = \frac{p(\mathbf{x}_t \mid x_0, \mathbf{x}_{t-1}, x_T)p(\mathbf{x}_{t-1} \mid \mathbf{x}_0, \mathbf{x}_T)}{p(\mathbf{x}_t \mid x_0, x_T)} \\
& = \frac{p(\mathbf{x}_t \mid \mathbf{x}_{t-1}, x_T)p(\mathbf{x}_{t-1} \mid x_0, \mathbf{x}_T)}{p(\mathbf{x}_t \mid x_0, x_T)}.
\end{aligned}
\tag{61}
$$

According to Appendix A.4, applying the reparameterization tricks:

$$
\begin{aligned}
\mathbf{x}_t & = e^{\bar{f}_t}\left(\frac{\gamma^{-1} + e^{2\bar{f}_T}\bar{g}_{t:T}^2}{\gamma^{-1} + e^{2\bar{f}_T}\bar{g}_T^2}x_0 + \frac{e^{\bar{f}_T}\bar{g}_t^2}{\gamma^{-1} + e^{2\bar{f}_T}\bar{g}_T^2}x_T + \left(\bar{h}_t - \frac{e^{2\bar{f}_T}\bar{h}_T\bar{g}_t^2}{\gamma^{-1} + e^{2\bar{f}_T}\bar{g}_T^2}\right)\mathbf{m}\right) + \bar{\sigma}_t'\epsilon_t \\
& \triangleq a_{t,\gamma}x_0 + b_{t,\gamma}x_T + c_{t,\gamma}\mathbf{m} + \bar{\sigma}_t'\epsilon_t, \\
\mathbf{x}_{t-1} & = e^{\bar{f}_{t-1}}\left(\frac{\gamma^{-1} + e^{2\bar{f}_T}\bar{g}_{t-1:T}^2}{\gamma^{-1} + e^{2\bar{f}_T}\bar{g}_T^2}x_0 + \frac{e^{\bar{f}_T}\bar{g}_{t-1}^2}{\gamma^{-1} + e^{2\bar{f}_T}\bar{g}_T^2}x_T + \left(\bar{h}_{t-1} - \frac{e^{2\bar{f}_T}\bar{h}_T\bar{g}_{t-1}^2}{\gamma^{-1} + e^{2\bar{f}_T}\bar{g}_T^2}\right)\mathbf{m}\right) + \bar{\sigma}_{t-1}'\epsilon_{t-1} \\
& = a_{t-1,\gamma}x_0 + b_{t-1,\gamma}x_T + c_{t-1,\gamma}\mathbf{m} + \bar{\sigma}_{t-1}'\epsilon_{t-1}.
\end{aligned}
\tag{62}
$$

Therefore, eliminating $x_0$ to obtain the relationships between $\mathbf{x}_t$, $\mathbf{x}_{t-1}$, $x_T$, $\mathbf{m}$ and noise $\epsilon$,

$$
\Rightarrow \quad \mathbf{x}_t = \frac{a_{t,\gamma}}{a_{t-1,\gamma}}\mathbf{x}_{t-1} + \left(b_{t,\gamma} - b_{t-1,\gamma}\frac{a_{t,\gamma}}{a_{t-1,\gamma}}\right)x_T + \left(c_{t,\gamma} - c_{t-1,\gamma}\frac{a_{t,\gamma}}{a_{t-1,\gamma}}\right)\mathbf{m} + \sqrt{\bar{\sigma}_t'^2 - \bar{\sigma}_{t-1}'^2\frac{a_{t,\gamma}^2}{a_{t-1,\gamma}^2}}\epsilon.
\tag{63}
$$

The mean value $\boldsymbol{\mu}_{t-1,\gamma}$ of $p(\mathbf{x}_{t-1} \mid x_0, \mathbf{x}_t, x_T)$ can be calculated as:

$$
\boldsymbol{\mu}_{t-1,\gamma} = \frac{\bar{\sigma}_{t-1}'^2 \frac{a_{t,\gamma}}{a_{t-1,\gamma}} \left[ \mathbf{x}_t - \left( b_{t,\gamma} - b_{t-1,\gamma} \frac{a_{t,\gamma}}{a_{t-1,\gamma}} \right) x_T - \left( c_{t,\gamma} - c_{t-1,\gamma} \frac{a_{t,\gamma}}{a_{t-1,\gamma}} \right) \mathbf{m} \right] + \left( \bar{\sigma}_t'^2 - \bar{\sigma}_{t-1}'^2 \frac{a_{t,\gamma}^2}{a_{t-1,\gamma}^2} \right) \bar{\boldsymbol{\mu}}_{t-1,\gamma}}{\bar{\sigma}_t'^2}
$$

$$
= \bar{\boldsymbol{\mu}}_{t-1,\gamma} - \frac{a_{t,\gamma}^2 \bar{\sigma}_{t-1}'^2}{a_{t-1,\gamma}^2 \bar{\sigma}_t'^2} \bar{\boldsymbol{\mu}}_{t-1,\gamma} + \frac{a_{t,\gamma} \bar{\sigma}_{t-1}'^2}{a_{t-1,\gamma} \bar{\sigma}_t'^2} \left[ \mathbf{x}_t - \left( b_{t,\gamma} - \frac{a_{t,\gamma} b_{t-1,\gamma}}{a_{t-1,\gamma}} \right) x_T - \left( c_{t,\gamma} - \frac{a_{t,\gamma} c_{t-1,\gamma}}{a_{t-1,\gamma}} \right) \mathbf{m} \right]
$$

$$
= \bar{\boldsymbol{\mu}}_{t-1,\gamma} + \frac{a_{t,\gamma} \bar{\sigma}_{t-1}'^2}{a_{t-1,\gamma} \bar{\sigma}_t'^2} \mathbf{x}_t - \frac{a_{t,\gamma} \bar{\sigma}_{t-1}'^2}{a_{t-1,\gamma} \bar{\sigma}_t'^2} \bar{\boldsymbol{\mu}}_{t,\gamma}
$$

$$
= \bar{\boldsymbol{\mu}}_{t-1,\gamma} + \frac{\bar{\sigma}_{t-1}'^2 a_{t,\gamma}}{\bar{\sigma}_t'^2 a_{t-1,\gamma}} (\mathbf{x}_t - \bar{\boldsymbol{\mu}}_{t,\gamma}).
$$

$(64)$

with the fact that

$$
\bar{\boldsymbol{\mu}}_{t,\gamma} = \frac{a_{t,\gamma}}{a_{t-1,\gamma}} \bar{\boldsymbol{\mu}}_{t-1,\gamma} + \left( b_{t,\gamma} - \frac{a_{t,\gamma} b_{t-1,\gamma}}{a_{t-1,\gamma}} \right) x_T + \left( c_{t,\gamma} - \frac{a_{t,\gamma} c_{t-1,\gamma}}{a_{t-1,\gamma}} \right) \mathbf{m},
\tag{65}
$$

which can be easily proved by expanding and comparing the both sides of the equation.

As for $\boldsymbol{\mu}_{\theta,t-1}$ and $\sigma_{t-1,\theta}^2$, parameterized from the SDE (15):

$$
\mathbf{x}_{t-1} = \mathbf{x}_t - \left[ f_t \mathbf{x}_t + h_t \mathbf{m} + g_t^2 \frac{x_T - e^{\bar{f}_{t:T}} \mathbf{x}_t - \mathbf{m} e^{\bar{f}_T} \bar{h}_{t:T}}{e^{-\bar{f}_{t:T}} (\gamma^{-1} + e^{2\bar{f}_T} \bar{g}_{t:T}^2)} - g_t^2 \nabla_{\mathbf{x}_t} \log p(\mathbf{x}_t \mid x_T) \right] - g_t \epsilon_t
$$

$$
\approx \mathbf{x}_t - \left[ f_t \mathbf{x}_t + h_t \mathbf{m} + g_t^2 \frac{x_T - e^{\bar{f}_{t:T}} \mathbf{x}_t - \mathbf{m} e^{\bar{f}_T} \bar{h}_{t:T}}{e^{-\bar{f}_{t:T}} (\gamma^{-1} + e^{2\bar{f}_T} \bar{g}_{t:T}^2)} - \frac{g_t^2}{\bar{\sigma}_t'} \boldsymbol{\epsilon}_\theta(\mathbf{x}_t, x_T, t) \right] - g_t \epsilon_t,
$$

$(66)$

where $\epsilon_t \sim N(\mathbf{0}, dt\boldsymbol{I})$.

Hence,

$$
\boldsymbol{\mu}_{\theta,t-1} = \mathbf{x}_t - \left[ f_t \mathbf{x}_t + h_t \mathbf{m} + g_t^2 \frac{x_T - e^{\bar{f}_{t:T}} \mathbf{x}_t - \mathbf{m} e^{\bar{f}_T} \bar{h}_{t:T}}{e^{-\bar{f}_{t:T}} (\gamma^{-1} + e^{2\bar{f}_T} \bar{g}_{t:T}^2)} - \frac{g_t^2}{\bar{\sigma}_t'} \boldsymbol{\epsilon}_\theta(\mathbf{x}_t, x_T, t) \right],
$$

$$
\sigma_{\theta,t-1} = g_t,
$$

$(67)$

which concludes the derivation of the training objective (19). $\qquad\square$

## A.6. Proof of Proposition 4.4

**Proposition 4.4.** *UniDB encompasses existing diffusion bridge models by employing different hyper-parameter spaces $\mathcal{H}$ as follows:*

- *DDBMs (VE) corresponds to UniDB with hyper-parameter $\mathcal{H}_{\text{VE}}(f_t = 0, h_t = 0, \gamma \to \infty)$*

- *DDBMs (VP) corresponds to UniDB with hyper-parameter $\mathcal{H}_{\text{VP}}(f_t = -\frac{1}{2} g_t^2, h_t = 0, \gamma \to \infty)$*

- *GOUB corresponds to UniDB with hyper-parameter $\mathcal{H}_{\text{GOU}}(f_t = \theta_t, h_t = -\theta_t, \mathbf{m} = \boldsymbol{\mu}, \gamma \to \infty)$*

*Proof.* **DDBMs (VE).**

$$
\mathcal{H}_{\text{VE}}(f_t = 0, h_t = 0, \gamma \to \infty)
$$

$$
\Leftrightarrow \quad \text{SOC problem with SDE: } d\mathbf{x}_t = g_t d\mathbf{w}_t \text{ as } \gamma \to \infty
$$

$$
\Leftrightarrow \quad d\mathbf{x}_t = \frac{\mathbf{x}_T - \mathbf{x}_t}{\int_t^T g_z^2 dz} dt + g_t d\mathbf{w}_t
$$

$$
\Leftrightarrow \quad d\mathbf{x}_t = \frac{\mathbf{x}_T - \mathbf{x}_t}{\sigma_T^2 - \sigma_t^2} dt + g_t d\mathbf{w}_t \text{ with } g_t^2 = \frac{d}{dt} \sigma_t^2
$$

$$
\Leftrightarrow \quad \text{DDBMs (VE) with Doob's } h\text{-transform}
$$

**DDBMs (VP).**

$$\mathcal{H}_{\text{VP}}(f_t = -\frac{1}{2}g_t^2, h_t = 0, \gamma \to \infty)$$

$\Leftrightarrow$ SOC problem with SDE: $\mathrm{d}\mathbf{x}_t = -\frac{1}{2}g_t^2\mathbf{x}_t\mathrm{d}t + g_t\mathrm{d}\mathbf{w}_t$ as $\gamma \to \infty$

$\Leftrightarrow$ $\mathrm{d}\mathbf{x}_t = \left(-\frac{1}{2}g_t^2\mathbf{x}_t + g_t^2 e^{\int_0^t \frac{g_z^2}{2}dz} \dfrac{e^{-\frac{1}{2}\int_0^t g_z^2 dz}\mathbf{x}_T - e^{-\frac{1}{2}\int_0^T g_z^2 dz}\mathbf{x}_t}{e^{\frac{1}{2}\int_t^T g_z^2 dz} - e^{-\frac{1}{2}\int_t^T g_z^2 dz}}\right)\mathrm{d}t + g_t\mathrm{d}\mathbf{w}_t$

$\Leftrightarrow$ $\mathrm{d}\mathbf{x}_t = \left(-\frac{1}{2}g_t^2\mathbf{x}_t + g_t^2 \dfrac{\alpha_t\mathbf{x}_T - \alpha_T\mathbf{x}_t}{\frac{\alpha_t^2\sigma_T^2}{\alpha_T} - \sigma_t^2\alpha_T}\right)\mathrm{d}t + g_t\mathrm{d}\mathbf{w}_t$ where $\alpha_t = e^{-\frac{1}{2}\int_0^t g_z^2 dz}$ and $g_t^2 = \frac{d}{dt}\sigma_t^2 + g_t^2\sigma_t^2$

$\Leftrightarrow$ $\mathrm{d}\mathbf{x}_t = \left(-\frac{1}{2}g_t^2\mathbf{x}_t + g_t^2 \dfrac{\frac{\alpha_t}{\alpha_T}\mathbf{x}_T - \mathbf{x}_t}{\sigma_t^2(\frac{\text{SNR}_t}{\text{SNR}_T} - 1)}\right)\mathrm{d}t + g_t\mathrm{d}\mathbf{w}_t$ where $\text{SNR}_t = \dfrac{\alpha_t^2}{\sigma_t^2}$

$\Leftrightarrow$ DDBMs (VP) with Doob's $h$-transform

**GOUB.**

$$\mathcal{H}_{\text{GOU}}(f_t = \theta_t, h_t = -\theta_t, \mathbf{m} = \boldsymbol{\mu}, \gamma \to \infty)$$

$\Leftrightarrow$ SOC problem with SDE: $\mathrm{d}\mathbf{x}_t = \theta_t(\boldsymbol{\mu} - \mathbf{x}_t)\mathrm{d}t + g_t\mathrm{d}\mathbf{w}_t$ as $\gamma \to \infty$

$\Leftrightarrow$ $\mathrm{d}\mathbf{x}_t = \left(\theta_t + g_t^2\dfrac{e^{-2\bar{\theta}_{t:T}}}{\bar{\sigma}_{t:T}^2}\right)(x_T - \mathbf{x}_t)\mathrm{d}t + g_t\mathrm{d}\mathbf{w}_t$

$\Leftrightarrow$ GOUB with Doob's $h$-transform

which concludes the proof of the Proposition 4.4.

$\square$

### A.7. Derivation of UniDB-GOU (forward SDE (20) and mean value of forward transition (21))

*Consider the SOC problem with GOU process (5), the optimally-controlled forward SDE is*

$$\mathrm{d}\mathbf{x}_t = \left(\theta_t + g_t^2\frac{e^{-2\bar{\theta}_{t:T}}}{\gamma^{-1} + \bar{\sigma}_{t:T}^2}\right)(x_T - \mathbf{x}_t)\mathrm{d}t + g_t\mathrm{d}\mathbf{w}_t, \tag{20}$$

*and the mean value of the probability $p(\mathbf{x}_t \mid x_0, x_T)$ is*

$$\bar{\boldsymbol{\mu}}_{t,\gamma} = e^{-\bar{\theta}_t}\frac{1 + \gamma\bar{\sigma}_{t:T}^2}{1 + \gamma\bar{\sigma}_T^2}x_0 + \left(1 - e^{-\bar{\theta}_t}\frac{1 + \gamma\bar{\sigma}_{t:T}^2}{1 + \gamma\bar{\sigma}_T^2}\right)x_T. \tag{21}$$

*Proof.* Consider the SOC problem with GOU process (5) in the deterministic form:

$$\min_{\mathbf{u}_{t,\gamma}} \int_0^T \frac{1}{2}\|\mathbf{u}_{t,\gamma}\|_2^2 dt + \frac{\gamma}{2}\|\mathbf{x}_T^u - x_T\|_2^2 \tag{68}$$
$$\text{s.t.} \quad \mathrm{d}\mathbf{x}_t = (\theta_t(x_T - \mathbf{x}_t) + g_t\mathbf{u}_{t,\gamma})\,\mathrm{d}t, \quad \mathbf{x}_0 = x_0$$

where the definition of $\boldsymbol{\mu}$ and $g_t$ is the same as GOUB: $\boldsymbol{\mu} = x_T$ $g_t^2 = 2\lambda^2\theta_t$.

Similarly to the proof of Proposition A.1, according to minimum principle theorem to obtain the following set of differential equations:

$$\frac{\mathrm{d}\mathbf{x}_t}{\mathrm{d}t} = \nabla_{\mathbf{p}_t} H\left(\mathbf{x}_t, \mathbf{p}_t, \mathbf{u}_{t,\gamma}^*, t\right) = \theta_t x_T - \theta_t\mathbf{x}_t - g_t^2\mathbf{p}_t, \tag{69}$$

$$\frac{d\mathbf{p}_t}{dt} = -\nabla_{\mathbf{x}_t} H\left(\mathbf{x}_t, \mathbf{p}_t, \mathbf{u}^*_{t,\gamma}, t\right) = \theta_t \mathbf{p}_t, \tag{70}$$

$$\mathbf{x}_0 = x_0, \tag{71}$$

$$\mathbf{p}_T = \gamma\left(\mathbf{x}_T - x_T\right). \tag{72}$$

Solving the equation (70), we have:

$$\begin{aligned}
\mathbf{p}_t &= \mathbf{p}_0 e^{\bar{\theta}_t}, \\
\mathbf{p}_T &= \mathbf{p}_0 e^{\bar{\theta}_T},
\end{aligned} \tag{73}$$

Then we solve the equation (69):

$$\begin{aligned}
& \frac{d\mathbf{x}_t}{dt} = \theta_t x_T - \theta_t \mathbf{x}_t - g_t^2 \mathbf{p}_t \\
\Rightarrow\quad & \frac{d(e^{\bar{\theta}_t}\mathbf{x}_t)}{dt} = e^{\bar{\theta}_t}\theta_t x_T - e^{\bar{\theta}_t}g_t^2 \mathbf{p}_t, \\
\Rightarrow\quad & e^{\bar{\theta}_t}\mathbf{x}_t - \mathbf{x}_0 = x_T \int_0^t e^{\bar{\theta}_z}\theta_z dz - \mathbf{p}_0 \int_0^t g_z^2 e^{2\bar{\theta}_z}dz, \\
\Rightarrow\quad & e^{\bar{\theta}_t}\mathbf{x}_t - x_0 = x_T(e^{\bar{\theta}_t}-1) - \lambda^2 \mathbf{p}_0 (e^{2\bar{\theta}_t}-1).
\end{aligned}$$

Hence, we can get:

$$\mathbf{x}_T = e^{-\bar{\theta}_T}x_0 + (1-e^{-\bar{\theta}_T})x_T - \lambda^2 \mathbf{p}_T(1-e^{-2\bar{\theta}_T}), \tag{74}$$

and

$$\mathbf{x}_t = e^{-\bar{\theta}_t}x_0 + (1-e^{-\bar{\theta}_t})x_T - \lambda^2 e^{-\bar{\theta}_T}\mathbf{p}_T(e^{\bar{\theta}_t}-e^{-\bar{\theta}_t}). \tag{75}$$

Take the equation (74) into the equation (72) and solve $\mathbf{p}_T$,

$$\begin{aligned}
& \mathbf{p}_T = \gamma\left(e^{-\bar{\theta}_1}x_0 + (1-e^{-\bar{\theta}_T})x_T - \lambda^2 \mathbf{p}_T(1-e^{-2\bar{\theta}_T}) - x_T\right) \\
\Rightarrow\quad & \mathbf{p}_T = \frac{\gamma e^{-\bar{\theta}_T}(x_0 - x_T)}{1+\gamma\lambda^2(1-e^{-2\bar{\theta}_T})}.
\end{aligned}$$

Hence,

$$\begin{aligned}
\mathbf{x}_t &= e^{-\bar{\theta}_t}x_0 + (1-e^{-\bar{\theta}_t})x_T - \lambda^2 e^{-\bar{\theta}_T}(e^{\bar{\theta}_t}-e^{-\bar{\theta}_t})\frac{\gamma e^{-\bar{\theta}_T}(x_0 - x_T)}{1+\gamma\lambda^2(1-e^{-2\bar{\theta}_T})} \\
&= \left(e^{-\bar{\theta}_t} - \frac{\gamma\lambda^2 e^{-2\bar{\theta}_T}(e^{\bar{\theta}_t}-e^{-\bar{\theta}_t})}{1+\gamma\lambda^2(1-e^{-2\bar{\theta}_T})}\right)x_0 + \left(1-e^{-\bar{\theta}_t}+\frac{\gamma\lambda^2 e^{-2\bar{\theta}_T}(e^{\bar{\theta}_t}-e^{-\bar{\theta}_t})}{1+\gamma\lambda^2(1-e^{-2\bar{\theta}_T})}\right)x_T \\
&= \left(e^{-\bar{\theta}_t}\frac{1+\gamma\lambda^2(1-e^{-2\bar{\theta}_{t:T}})}{1+\gamma\lambda^2(1-e^{-2\bar{\theta}_T})}\right)x_0 + \left(1-e^{-\bar{\theta}_t}\frac{1+\gamma\lambda^2(1-e^{-2\bar{\theta}_{t:T}})}{1+\gamma\lambda^2(1-e^{-2\bar{\theta}_1})}\right)x_T \\
&= e^{-\bar{\theta}_t}\frac{1+\gamma\bar{\sigma}^2_{t:T}}{1+\gamma\bar{\sigma}^2_T}x_0 + \left(1-e^{-\bar{\theta}_t}\frac{1+\gamma\bar{\sigma}^2_{t:T}}{1+\gamma\bar{\sigma}^2_T}\right)x_T,
\end{aligned} \tag{76}$$

which implies

$$\bar{\boldsymbol{\mu}}_{t,\gamma} = e^{-\bar{\theta}_t}\frac{1+\gamma\bar{\sigma}^2_{t:T}}{1+\gamma\bar{\sigma}^2_T}x_0 + \left(1-e^{-\bar{\theta}_t}\frac{1+\gamma\bar{\sigma}^2_{t:T}}{1+\gamma\bar{\sigma}^2_T}\right)x_T. \tag{77}$$

Then,

$$
\begin{aligned}
\mathbf{u}_{t,\gamma}^* &= -g_t \mathbf{p}_t \\
&= -g_t e^{\bar{\theta}_t} e^{-\bar{\theta}_T} \frac{\gamma e^{-\bar{\theta}_T}(x_0 - x_T)}{1 + \gamma\lambda^2(1 - e^{-2\bar{\theta}_T})} \\
&= -g_t e^{\bar{\theta}_t} e^{-\bar{\theta}_T} \frac{\gamma e^{-\bar{\theta}_T}(x_0 - x_T)}{1 + \gamma\bar{\sigma}_T^2} \\
&= -g_t e^{\bar{\theta}_t} e^{-\bar{\theta}_T} \frac{\gamma e^{-\bar{\theta}_T} e^{\bar{\theta}_t}(\mathbf{x}_t - x_T)}{1 + \gamma\bar{\sigma}_{t:T}^2} \\
&= g_t \frac{e^{-2\bar{\theta}_{t:T}}(x_T - \mathbf{x}_t)}{\gamma^{-1} + \bar{\sigma}_{t:T}^2}.
\end{aligned}
\tag{78}
$$

And the optimally-controlled dynamics can be:

$$
\mathrm{d}\mathbf{x}_t = \left( \theta_t + g_t^2 \frac{e^{-2\bar{\theta}_{t:T}}}{\gamma^{-1} + \bar{\sigma}_{t:T}^2} \right)(x_T - \mathbf{x}_t)\mathrm{d}t + g_t \mathrm{d}\mathbf{w}_t,
\tag{79}
$$

which concludes the derivation of UniDB-GOU (forward SDE (20) and mean value of forward transition (21)).  □

### A.8. Examples of UniDB-VE and UniDB-VP

Similar to section 4.5, we provide the other examples of UniDB-VE and UniDB-VP, highlighting the key difference of the coefficient of $x_0$ in the mean value of forward transition and $h$-function term between UniDB and them respectively.

**UniDB-VE**

$$
\begin{aligned}
\frac{\sigma_T^2 - \sigma_t^2}{\sigma_T^2 - \sigma_0^2} &\Rightarrow \frac{\gamma^{-1} + \sigma_T^2 - \sigma_t^2}{\gamma^{-1} + \sigma_T^2 - \sigma_0^2} \\
\underbrace{\mathbf{h} = \frac{x_T - \mathbf{x}_t}{\sigma_T^2 - \sigma_t^2}}_{\text{VE}} &\Rightarrow \underbrace{\frac{\mathbf{u}_{t,\gamma}^*}{g_t} = \frac{x_T - \mathbf{x}_t}{\gamma^{-1} + \sigma_T^2 - \sigma_t^2}}_{\text{UniDB-VE}}
\end{aligned}
\tag{80}
$$

**UniDB-VP**

$$
\begin{aligned}
\alpha_t\left(1 - \frac{\mathrm{SNR}_T}{\mathrm{SNR}_t}\right) &\Rightarrow \alpha_t\left(1 - \frac{\frac{\alpha_t^2 \alpha_T}{\sigma_t^2 \sigma_T^2}\gamma^{-1} + \mathrm{SNR}_T}{\frac{\alpha_t^2 \alpha_T}{\sigma_t^2 \sigma_T^2}\gamma^{-1} + \mathrm{SNR}_t}\right) \\
\underbrace{\mathbf{h} = \frac{\frac{\alpha_t}{\alpha_T}x_T - \mathbf{x}_t}{\sigma_t^2\left(\frac{\mathrm{SNR}_t}{\mathrm{SNR}_T} - 1\right)}}_{\text{VP}} &\Rightarrow \underbrace{\frac{\mathbf{u}_{t,\gamma}^*}{g_t} = \frac{\frac{\alpha_t}{\alpha_T}x_T - \mathbf{x}_t}{\gamma^{-1}\frac{\alpha_t}{\alpha_T} + \sigma_t^2\left(\frac{\mathrm{SNR}_t}{\mathrm{SNR}_T} - 1\right)}}_{\text{UniDB-VP}}
\end{aligned}
\tag{81}
$$

### A.9. Proof of Proposition 4.5

**Proposition 4.5.** *Denote the initial state distribution $x_0$, the terminal distribution $\mathbf{x}_T^u$ by the controller and the pre-defined terminal distribution $x_T$, then*

$$
\|\mathbf{x}_T^u - x_T\|_2^2 = \frac{e^{-2\bar{\theta}_T}}{\left(1 + \gamma\lambda^2(1 - e^{-2\bar{\theta}_T})\right)^2}\|x_T - x_0\|_2^2.
\tag{23}
$$

*Proof.* According to Appendix A.7, we've learned that

$$
\mathbf{x}_t^u = e^{-\bar{\theta}_t}\frac{1 + \gamma\bar{\sigma}_{t:T}^2}{1 + \gamma\bar{\sigma}_T^2}x_0 + \left(1 - e^{-\bar{\theta}_t}\frac{1 + \gamma\bar{\sigma}_{t:T}^2}{1 + \gamma\bar{\sigma}_T^2}\right)x_T + \frac{\bar{\sigma}_t^2 \bar{\sigma}_{t:T}^2}{\bar{\sigma}_T^2}\epsilon.
\tag{82}
$$

Take $t = T$, then

$$
\mathbf{x}_T^u = \frac{e^{-\bar{\theta}_t}}{1 + \gamma\bar{\sigma}_T^2}x_0 + \left(1 - \frac{e^{-\bar{\theta}_t}}{1 + \gamma\bar{\sigma}_T^2}\right)x_T.
\tag{83}
$$

Therefore, since $\bar{\sigma}_T^2 = \lambda^2(1 - e^{-2\bar{\theta}_T})$,

$$\|\mathbf{x}_T^u - x_T\|_2^2 = \left\| \frac{e^{-\bar{\theta}_t}}{1 + \gamma\bar{\sigma}_T^2} x_0 + \left(1 - \frac{e^{-\bar{\theta}_t}}{1 + \gamma\bar{\sigma}_T^2}\right) x_T - x_T \right\|_2^2$$

$$= \frac{e^{-2\bar{\theta}_T}}{\left(1 + \gamma\lambda^2(1 - e^{-2\bar{\theta}_T})\right)^2} \|x_T - x_0\|_2^2, \tag{84}$$

which concludes the proof of the Proposition 4.5. $\qquad\square$

## B. Implementation Details

In Image Restoration Tasks (Image $4\times$Super-resolution, Image Deraining and Image Inpainting), we follow the experiment setting of GOUB (Yue et al., 2023): the same noise network which is similar to U-Net structure (Chung et al., 2022), steady variance level $\lambda^2 = 30^2/255^2$, coefficient $e^{\bar{\theta}_T} = 0.005$ instead of zero, sampling step number $T = 100$, 128 patch size with 8 batch size when training, Adam optimizer with $\beta_1 = 0.9$ and $\beta_2 = 0.99$ (Kingma, 2014), 1.2 million total training steps with $10^{-4}$ initial learning rate and decaying by half at 300, 500, 600, and 700 thousand iterations. With respect to the schedule of $\theta_t$, we choose a flipped version of cosine noise schedule (Nichol & Dhariwal, 2021; Luo et al., 2023),

$$\theta_t = 1 - \frac{cos(\frac{t/T+s}{1+s}\frac{\pi}{2})^2}{cos(\frac{s}{1+s}\frac{\pi}{2})^2} \tag{85}$$

where $s = 0.008$ to achieve a smooth noise schedule. $g_t$ is determined through $g_t^2 = 2\lambda^2\theta_t$. As for the datasets of the three main experiments, we take 800 images for training and 100 for testing for the DIV2K dataset, 1800 images for training and 100 for testing for the Rain100H dataset, 27000 images for training and 3000 for testing for the CelebA-HQ $256\times256$ dataset. Our models are trained on a single NVIDIA H20 GPU with 96GB memory for about 2 days.

# C. Additional Experimental Results

Here we will illustrate more experimental results.

*Table 3.* Qualitative comparison between different bridge models (DDBMs (VE) and DDBMs (VP)) and ours (UniDB-VE and UniDB-VP) on DIV2K and Rain100H datasets.

| METHOD | Image 4× Super-Resolution | | | | Image Deraining | | | |
|---|---|---|---|---|---|---|---|---|
| | PSNR↑ | SSIM↑ | LPIPS↓ | FID↓ | PSNR↑ | SSIM↑ | LPIPS↓ | FID↓ |
| DDBMs (VE) | 23.34 | 0.4295 | 0.372 | 32.28 | 29.34 | 0.7654 | **0.185** | 43.22 |
| UniDB-VE | **23.84** | **0.4454** | **0.357** | **31.29** | **29.46** | **0.7671** | 0.185 | **42.57** |
| DDBMs (VP) | 22.11 | 0.4059 | 0.491 | 48.09 | 29.58 | 0.828 | 0.113 | 35.46 |
| UniDB-VP | **22.42** | **0.4097** | **0.486** | **44.52** | **30.11** | **0.8414** | **0.102** | **33.17** |

*Table 4.* **Image 4×Super-Resolution.** Qualitative evaluation of the CelebA-HQ 256×256 datasets with baselines.

| METHOD | Penalty Coefficient $\gamma$ | PSNR↑ | SSIM↑ | LPIPS↓ | FID↓ |
|---|---|---|---|---|---|
| DDBM (VE) | $\infty$ | **25.84** | **0.5099** | 0.381 | 67.98 |
| UniDB-VE | $1 \times 10^7$ | 25.37 | 0.504 | **0.265** | **53.98** |
| DDBM (VP) | $\infty$ | 27.13 | 0.6497 | 0.194 | 42.54 |
| UniDB-VP | $1 \times 10^7$ | **27.44** | **0.6631** | **0.174** | **42.06** |
| GOUB | $\infty$ | 28.63 | 0.7776 | 0.104 | 19.02 |
| UniDB-GOU | $1 \times 10^7$ | **28.70** | **0.7894** | **0.090** | **17.59** |

*Table 5.* **Image 4×Super-Resolution.** Qualitative evaluation of the FFHQ 256×256 dataset with baselines.

| METHOD | LPIPS↓ | FID↓ |
|---|---|---|
| DDRM | 0.339 | 59.57 |
| DPS | 0.214 | 39.35 |
| DDBM (VE) | 0.239 | 42.85 |
| DDBM (VP) | 0.177 | 39.63 |
| GOUB | 0.072 | 21.77 |
| UniDB-GOU | **0.069** | **20.24** |

*Table 6.* **Image Inpainting.** Qualitative comparison with the relevant baselines on CelebA-HQ with thick mask.

| METHOD | PSNR↑ | SSIM↑ | LPIPS↓ | FID↓ |
|---|---|---|---|---|
| DDRM | 19.48 | 0.8154 | 0.1487 | 26.24 |
| IRSDE | 21.12 | 0.8499 | 0.1046 | 11.12 |
| GOUB | 22.59 | **0.8573** | 0.0917 | 8.49 |
| UniDB-GOU | **23.02** | 0.8571 | **0.0884** | **7.46** |

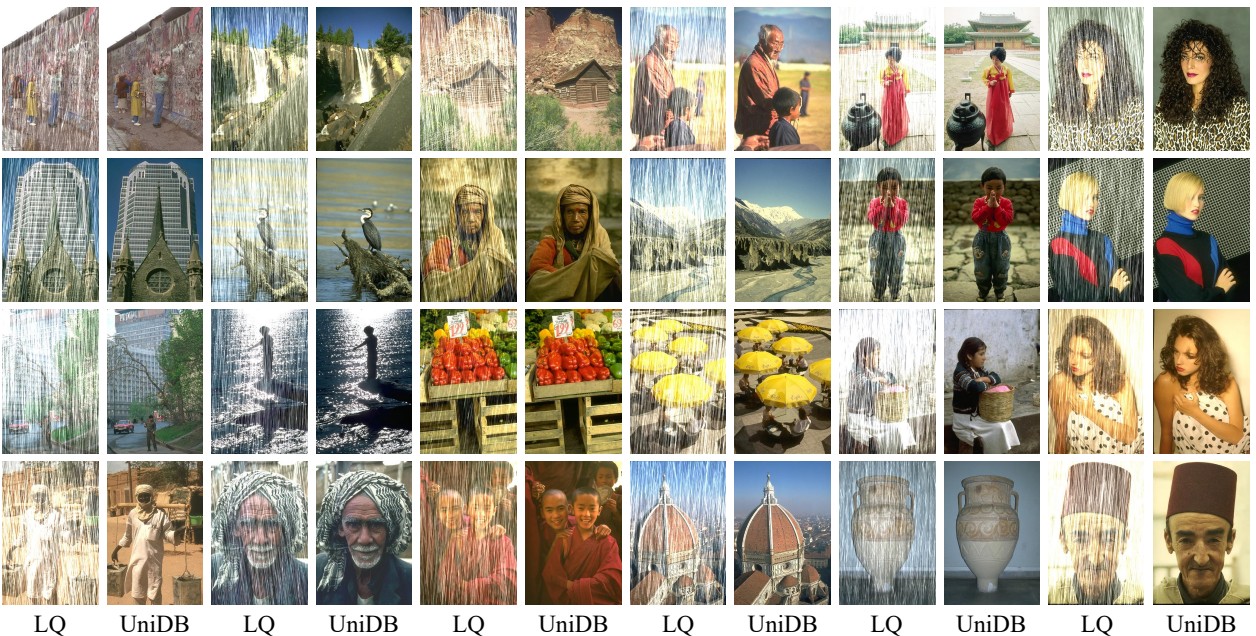

Figure 5. Additional visual results on deraining with Rain100H datasets.

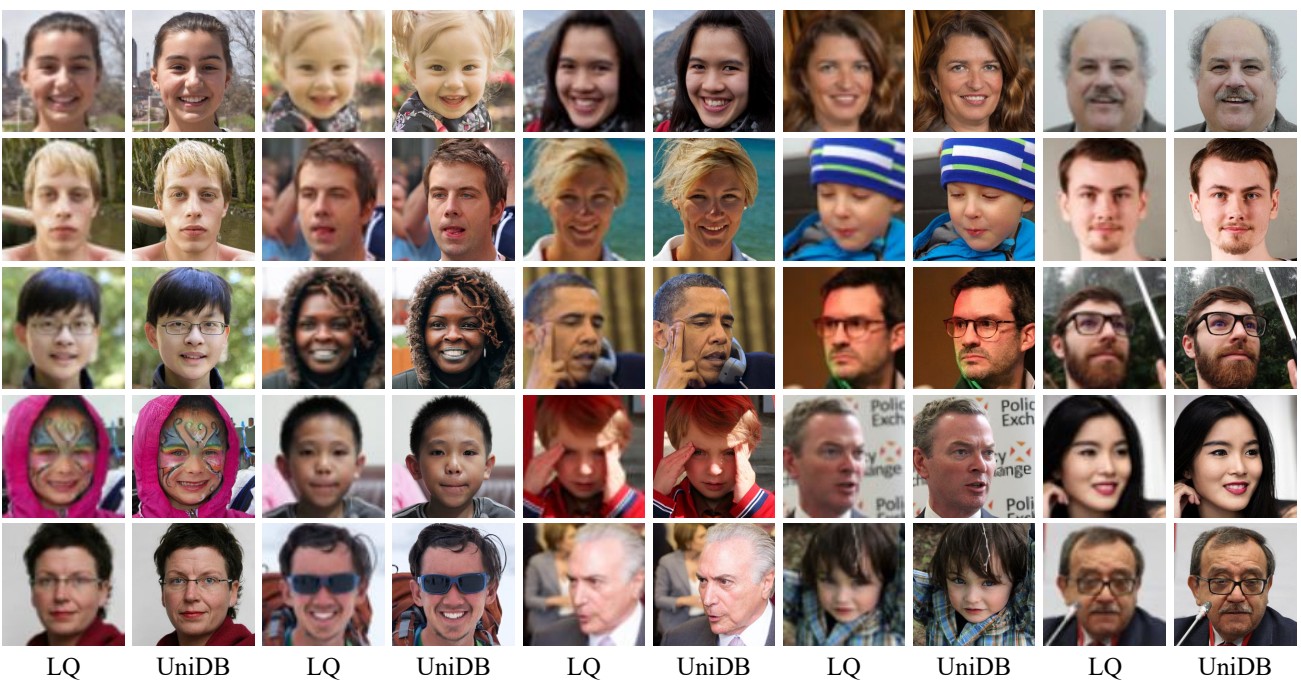

Figure 6. Additional visual results on 4×super-resolution with FFHQ datasets.

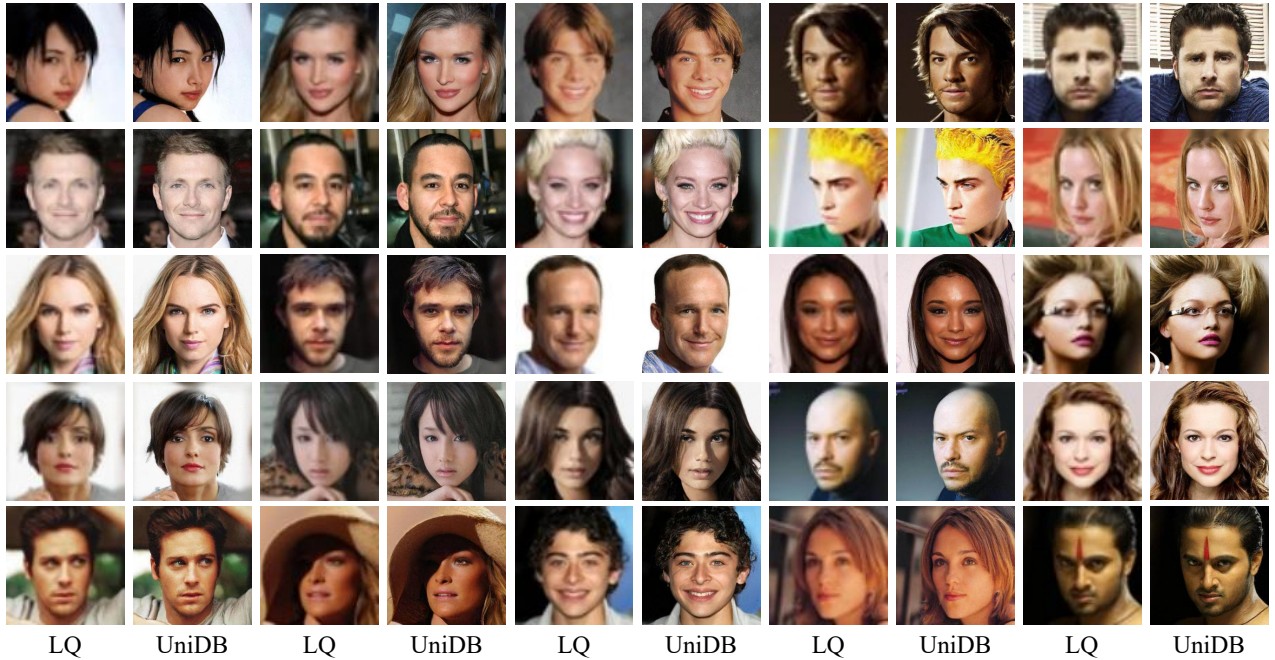

*Figure 7.* Additional visual results on 4×super-resolution with CelebA-HQ datasets.

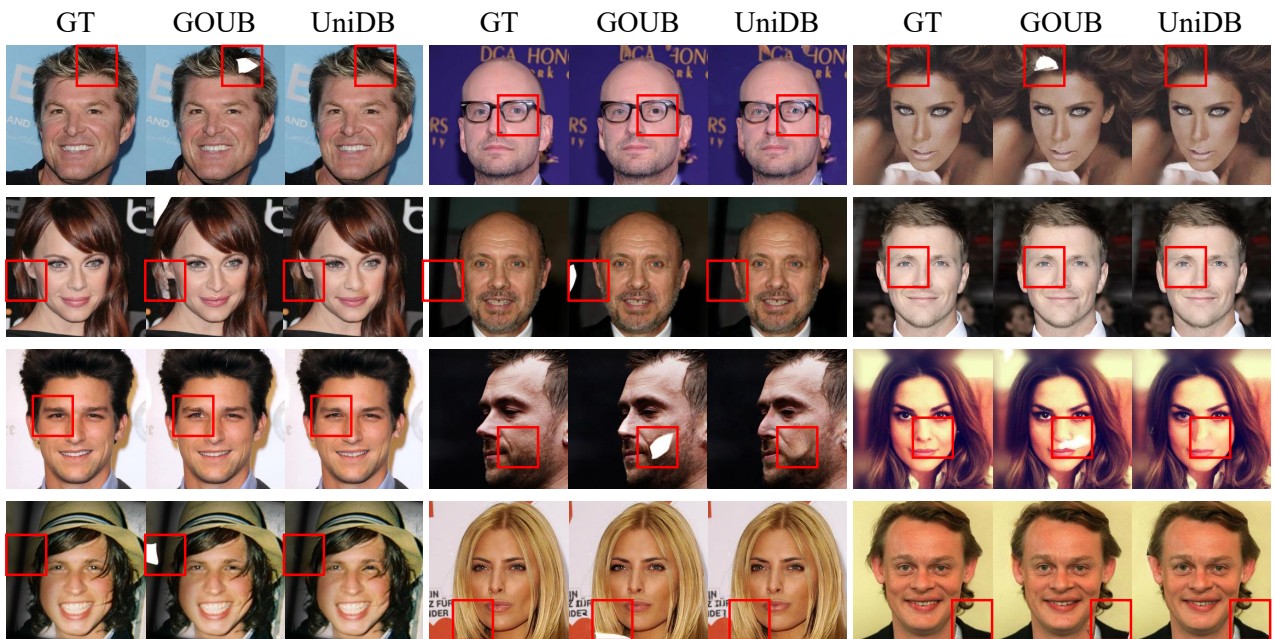

*Figure 8.* Additional visual results on thin mask inpainting with CelebA-HQ datasets to show our excellence.

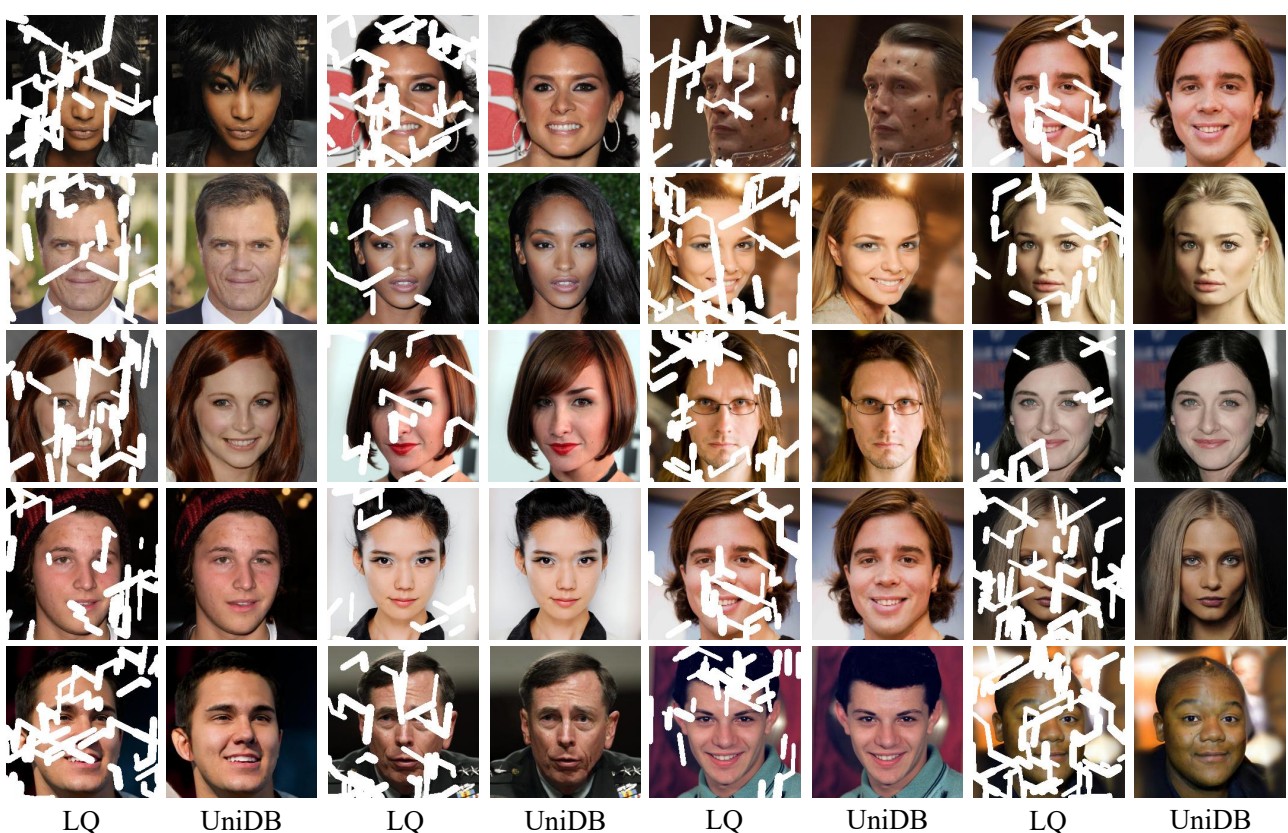

LQ    UniDB    LQ    UniDB    LQ    UniDB    LQ    UniDB

*Figure 9.* Additional visual results on thin mask inpainting with CelebA-HQ datasets.

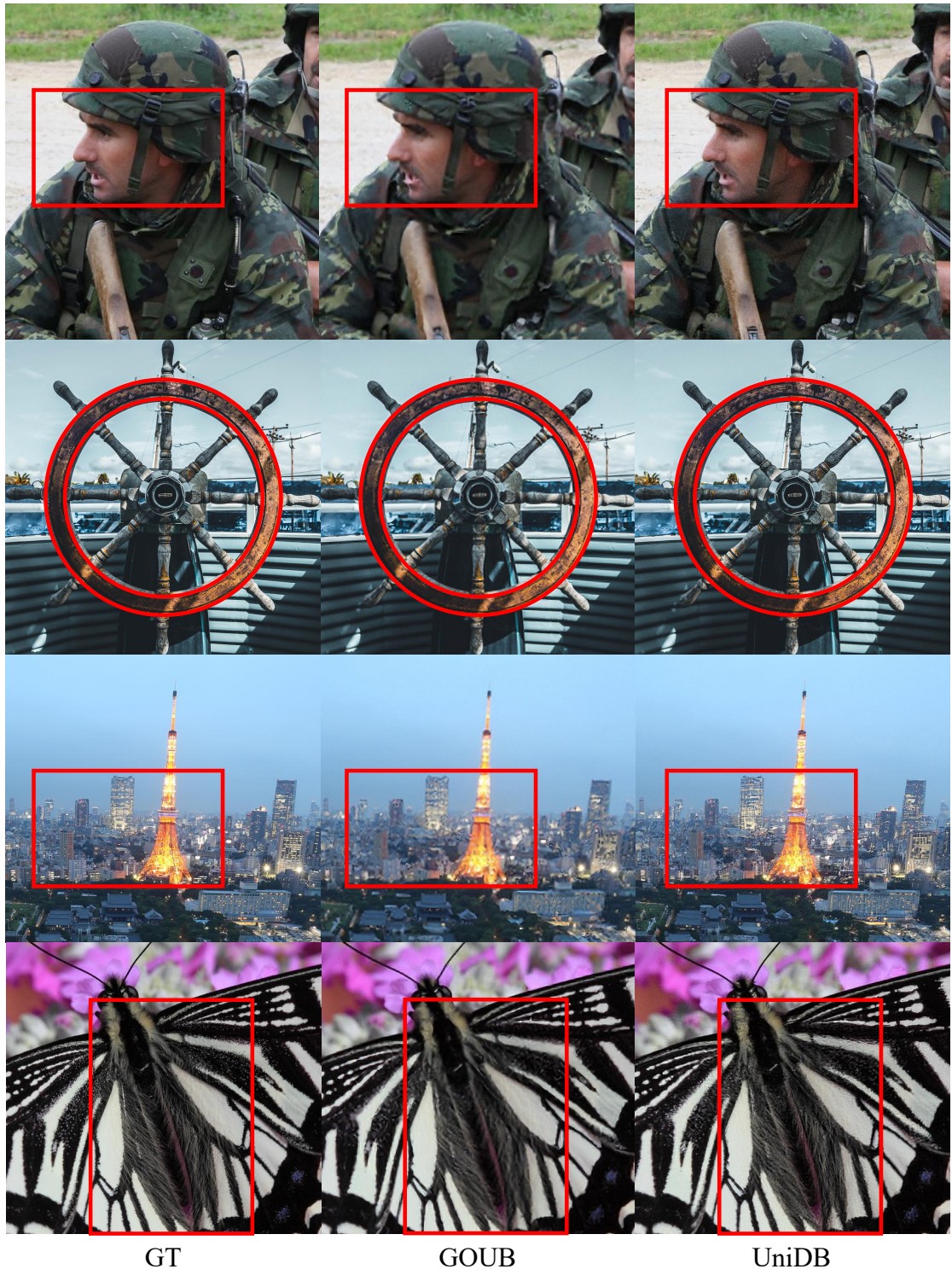

GT            GOUB            UniDB

*Figure 10.* Additional visual results on 4×super-resolution with DIV2K datasets.