# OpenReview forum: "UniDB: A Unified Diffusion Bridge Framework via Stochastic Optimal Control"
_ICML.cc/2025/Conference — ICML 2025 spotlightposter_

### Official Review · Reviewer_G2ra · 2025-03-10

**Overall Recommendation:** 4

**Summary:**

This paper proposes a unified framework, UniDB, of diffusion bridge models based on Stochastic Optimal Control. This framework enhances the quality and detail of generated images by balancing control cost and terminal penalty.

**Claims And Evidence:**

Claim 1: UniDB helps to understand and generalize Doob’s $h$-transform.
The theoretical derivations in the paper demonstrate that Doob’s $h$-transform is a special case of UniDB when the penalty coefficient approaches infinity. This provides a solid mathematical foundation for the claim.

Claim 2: UniDB improves image quality by allowing the design of different controllers $u_{t, \gamma}$.
The claim is supported by a lot of qualitative and quantified experimental results, demonstrating that controller design enhances image quality.

**Essential References Not Discussed:**

The key contribution of this paper is the introduction of Stochastic Optimal Control into the DDBM theoretical framework. However, a similar approach was explored last year in:

[1] Zhang, Shaorong, et al. "Exploring the Design Space of Diffusion Bridge Models via Stochasticity Control." arXiv preprint arXiv:2410.21553 (2024).

To differentiate from prior work, the paper should explicitly highlight the key distinctions between this work and Zhang et al. (2024), particularly in how Stochastic Optimal Control is formulated and applied.

**Experimental Designs Or Analyses:**

The experiments are comprehensive and well-designed, covering three different tasks to ensure robustness and generalizability. Additionally, the paper conducts an ablation study on the key penalty coefficient $\gamma$, which helps evaluate its impact on model performance.

However, I suggest that the authors include DDBM as a benchmark for comparison. Since DDBM is also a Doob’s $h$-transform-based model and is mentioned in the **Preliminaries**, it would be beneficial to compare the proposed method against it. This would provide a clearer assessment of the advantages and potential improvements offered by the proposed framework.

**Methods And Evaluation Criteria:**

The use of PSNR, SSIM, LPIPS, and FID as evaluation metrics in experiments on three high-resolution datasets (CelebA-HQ, Rain100H, and DIV2K) is reasonable. These metrics comprehensively assess image quality, covering aspects of pixel-wise similarity (PSNR, SSIM), perceptual quality (LPIPS), and generative diversity (FID). The chosen datasets are also well-suited for high-resolution image generation and restoration tasks, ensuring a robust evaluation.

**Other Comments Or Suggestions:**

See the above comments.

**Other Strengths And Weaknesses:**

The paper is well-structured and clearly organized, making it easy for readers to follow. However, the novelty is sceptical, as a similar approach was explored last year.

**Questions For Authors:**

See the above comments.

**Relation To Broader Scientific Literature:**

1. UniDB as a Generalization of Doob’s h-Transform:
Doob’s h-transform has been widely studied in stochastic processes and has been applied in bridge modeling [1][2]. The paper demonstrates that Doob’s $h$-transform is a special case of the proposed UniDB framework when the penalty coefficient tends to infinity, providing a broader theoretical foundation for diffusion bridge models.

2. Controller Design for Improved Image Generation:
Stochastic Optimal Control has been explored for diffusion bridge models [3], but existing approaches often lead to artifacts such as
blurred or distorted details. By allowing the design of different controllers $u_{t, \gamma}$, UniDB provides greater control over generation quality, leading to improved image fidelity and diversity across multiple datasets.

[1] Zhou, Linqi, et al. "Denoising diffusion bridge models.", 2023.

[2] Yue, C., et al. "Image restoration through generalized ornstein-uhlenbeck bridge", 2024.

[3] Park, B., et al. Stochastic optimal control for diffusion bridges in function spaces, 2024.

**Theoretical Claims:**

The mathematical derivations in the paper are complete and generally reliable. However, two aspects require further clarification from the authors:

1. Choice of $L_1$ norm in the training objective (Equation 19):
The paper does not provide a clear justification for using the $L_1$ norm instead of other alternatives like $L_2$ norm. The authors should explain whether this choice is based on empirical performance, theoretical considerations, or robustness to outliers.

2. Introduction of the state vector term $m$ in linear SDE:
One of the paper's key novelties is introducing the $m$ term in the linear SDE form, but it is not explicitly explained how it is computed or designed in the main context. Further elaboration on the motivation, computation, and impact of $m$ would improve the clarity of the contribution.

---

> ### Author Rebuttal · Authors · 2025-04-01
>
> Thank you for your feedbacks and comments.
>
> > Claim 1: Choice of $L_1$ norm in the training objective (Equation 19): The paper does not provide a clear justification for using the $L_1$ norm instead of other alternatives like $L_2$ norm. The authors should explain whether this choice is based on empirical performance, theoretical considerations, or robustness to outliers.
>
> Thanks for your feedback. **Using $L_1$ norm is a common trick** in the field of image restoration with **many prior works [G1-G4]**. We just follow to use the same trick as we mentioned in **line 270 of our paper**. The reason that choosing $L_1$ norm over other alternatives like $L_2$ norm in image restoration tasks have been analyzed in [G2-G4]. In [G2], "when the network is trained using $L_1$ as a cost function, instead of the traditional $L_2$, the average quality of the output images is superior for all the quality metrics considered." [G3, G4] establish the $L_1$ loss's superiority over $L_2$ through improved convergence and outlier robustness.
>
> [G1] Yue et al. "Image restoration through generalized ornstein-uhlenbeck bridge", ICML 2024.
>
> [G2] Zhao et al. "Loss Functions for Image Restoration with Neural Networks", IEEE Transactions on Computational Imaging 2017.
>
> [G3] Lim et al. "Enhanced Deep Residual Networks for Single Image Super-Resolution", CVPR 2017 workshop.
>
> [G4] Mu et al. "Riemannian Loss for Image Restoration", CVPR 2019 workshop.
>
> > Claim 2: Introduction of the state vector term $\mathbf{m}$ in linear SDE: One of the paper's key novelties is introducing the $\mathbf{m}$ term in the linear SDE form, but it is not explicitly explained how it is computed or designed in the main context. Further elaboration on the motivation, computation, and impact of $\mathbf{m}$ would improve the clarity of the contribution.
>
> We want to clarify that we've **never** claimed the introduction of the state vector term $\mathbf{m}$ in linear SDE is the **key novelty of our framework**. Instead, our main novelty lies in constructing the diffusion bridge in the form of stochastic optimal control and revealing that Doob's h-transform is a special case of ours. The introduction is just a **simple reformulation** referenced by prior works IR-SDE [G5] and GOUB [G1]: we reformulated the drift term $\theta_t(\mu - x_t)$ into $f_t x_t + h_t \mathbf{m}$. Particularly in our experiments of UniDB-GOU, for a fair comparison, we set $\mathbf{m} = \mu$ which is identical to GOUB [G1], ensuring consistency with baselines.
>
> [G5] Luo et al. "Image Restoration with Mean-Reverting Stochastic Differential Equations", ICML 2023.
>
> > Experimental Designs Or Analyses: The reviewer suggests that the authors should include DDBMs as a benchmark for comparison.
>
> Actually, we **have compared DDBMs** as a benchmark in **Appendix E "Additional Experimental Results"** of our paper, theoretically analyzing the application of UniDB to DDBMs (Appendix: A.8. Examples of UniDB-VE and UniDB-VP) and conducting extensive experiments (Tables 3, 4, and 5). Specifically, it outperforms DDBMs on both LPIPS and FID metrics, with gains reaching up to ~20\% in some cases.
>
> > Essential References Not Discussed: a similar approach was explored last year in: Zhang, Shaorong, et al. "Exploring the Design Space of Diffusion Bridge Models via Stochasticity Control". To differentiate from prior work, the paper should explicitly highlight the key distinctions between this work and Zhang et al. (2024), particularly in how Stochastic Optimal Control is formulated and applied.
>
> Our work takes a completely different approach from the prior work Zhang et al. (2024) (following simply denoted as SDB), specifically:
>    - **Different purposes.** SDB mainly focuses on solving **singularities and accelerations** for training and sampling caused by the neglection of the impact of noise in sampling SDEs. In contrast, our UniDB aims to **address the issues resulting from Doob's h-transform (e.g. artifacts along edges and unnatural patterns)** that occur in the existing diffusion bridge models.
>    - **Different methods.** Although both that paper and our work mention "Stochastic Control" in titles, **the two "Stochastic Control" are totally different**. SDB is more inclined towards **stochastic process**, adding noise into the base distribution and stochasticity to the reverse process which **is still based on Doob's h-transform**. While our UniDB focuses on stochastic **optimal** control, modeling the forward process as an **optimization problem** to analyze the drawbacks of Doob's h-transform.
>
> Therefore, the two articles are **fundamentally different** in nature.

---

> > ### Comment · Reviewer_G2ra · 2025-04-03
> >
> > Thanks for the author's clarification on my confusion. I've raised up my rating.

---

### Official Review · Reviewer_9hKM · 2025-03-11

**Overall Recommendation:** 4

**Summary:**

This paper introduces UniDB, a diffusion bridge model framework that utilizes
Stochastic Optimal Control (SOC) for process optimization, providing an analytical solution for the optimal controller. UniDB generalizes existing diffusion
bridge models by showing that Doob’s h-transform is a special case where the
penalty coefficient γ tends to infinity. By adjusting the trade-off between control
costs and terminal penalties, UniDB improves image detail and quality while
maintaining compatibility with existing models. Experimental results demonstrate its effectiveness in image restoration tasks.

## update after rebuttal
I have carefully read the rebuttal and would like to maintain my original score.

**Claims And Evidence:**

UniDB claims to generate higher-quality images than Doob’s h-transform and
supports this claim with experimental evidence on tasks such as super-resolution,
inpainting, and deraining. However, additional explanation is needed regarding
whether the optimal controller in LQ SOC directly contributes to producing
sharper and more detailed images.
Furthermore, through Proposition 4.3, it is shown from an LQ SOC perspective that the optimal controller obtained with a finite γ is preferable to that of
the infinite case. However, it remains unclear whether there is a systematic analysis regarding the choice of γ. In Proposition 4.5 and Figure 2, a sufficiently large
γ is selected to minimize differences in terminal point positions. Nevertheless,
based on the results in Table 2, it appears that using an optimal controller with
finite γ does not always lead to improvements in actual evaluation metrics. I
would appreciate further clarification on this point.

**Essential References Not Discussed:**

No.

**Experimental Designs Or Analyses:**

This paper follows a standard experimental design and evaluation process, so I
believe there are no issues in this regard.

**Methods And Evaluation Criteria:**

The authors demonstrate their approach’s strength through evaluation criteria
widely used in prior works

**Other Comments Or Suggestions:**

Please refer to the questions and comments provided above.

**Other Strengths And Weaknesses:**

Please refer to the questions and comments provided above.

**Questions For Authors:**

Please refer to the questions and comments provided above.

**Relation To Broader Scientific Literature:**

No.

**Theoretical Claims:**

The detailed proofs of the Theorems and Propositions in the main paper are clearly described in the Appendix. However, I noticed some minor issues when cross-referencing the statements with their proofs.

1. There are minor typos regarding the connection between the statements and the proofs. The authors are encouraged to check the appendix number references carefully.

2. In equation (53) of Appendix A.3, could the authors provide more details on the derivation of this part?
   $
    \frac{1}{2} \displaystyle\frac{\gamma}{(1+\gamma e ^ {2 \bar{f}_T} \bar{g}_T^2)^2} ||a||_2^2
    = \frac{\gamma}{2} || \mathbf{x}_T^u - x_T ||_2^2
    $
    The intermediate steps would help improve clarity.

---

> ### Author Rebuttal · Authors · 2025-04-01
>
> We greatly appreciate your feedback and inquiries.
>
> > Claims And Evidence 1: Additional explanation is needed regarding whether the optimal controller in LQ SOC directly contributes to producing sharper and more detailed images.
>
> The over-control in Doob's h-transform violates the natural statistical properties of images, **prioritizing mathematical precision (pixel-perfect endpoints) over visual authenticity (realistic SDE trajectories)**. UniDB leads to better overall performance considering **both realistic SDE trajectories and target endpoint matching**. A comprehensive analysis can be found in response for **Reviewer 33jh**'s
> Claim and response for **Reviewer AzN7**'s Claim and Question 2.
>
> > Claims And Evidence 2: It remains unclear whether there is a systematic analysis regarding the choice of $\gamma$. Based on the results in Table 2, it appears that using an optimal controller with finite $\gamma$ does not always lead to improvements in actual evaluation metrics.
>
> In **Figure 2** of our paper, we illustrate that a range of $10^5$ to $10^9$ for $\gamma$ is pretty good. What we want to emphasize is not that there is a single best value of $\gamma$ for each dataset, but rather that most value of $\gamma$ chosen **within this interval can yield a better result than Doob's h-transform ($\gamma = \infty$)**. While suboptimal $\gamma$ values may occasionally degrade performance, our approach provides **a simple, efficient, and nearly cost-free way** to improve model performance, which works well in most cases.
>
> It is worth noting that when $\gamma$ is set to $1e7$, our model **outperforms all baselines across multiple tasks**, including super-resolution (on DIV2K, CelebA, and FFHQ), deraining on (Rain100H), and inpainting on (CelebA).
>
> > Question: Could the authors provide more details on the derivation of this part: $\frac{\gamma}{2}\left\||\mathbf{x}_T^u-x_T\right\||_2^2=\frac{\gamma\||a\||_2^2}{2\left(1+\gamma e^{2 \bar{f}_T} \bar{g}_T^2\right)^2}$?
>
> Yes, below is the detailed proof: At line 770 of our paper, we defined $a=e^{\bar{f}_T}x_0-x_T+\mathbf{m}e^{\bar{f}_T}\bar{h}_T$ for simple denotation. The first equation of Eq. 51 is a simple simplification by substituting the expression $\mathbf{x}_T^u$ from Eq. 50 into Eq. 51. A more detailed proof is as follows:
>
> $\left\||\mathbf{x}_T^u-x_T\right\||_2^2=\left\|\left\|\left(\frac{\gamma^{-1} e^{f_T}}{\gamma^{-1}+e^{2 f_T} \bar{g}_T^2}\right) x_0+\left(\frac{e^{2 \bar{f}_T} \bar{g}_T^2}{\gamma^{-1}+e^{2 \bar{f}_T} \bar{g}_T^2}\right) x_T+e^{\bar{f}_T}\left(\frac{\gamma^{-1} \bar{h}_T}{\gamma^{-1}+e^{2 f_T} \bar{g}_T^2}\right) \mathbf{m}-x_T\right\|\right\|_2^2$
>
> $=\left\|\left\|\left(\frac{\gamma^{-1} e^{\bar{f}_T}}{\gamma^{-1}+e^{2 \bar{f}_T} \bar{g}_T^2}\right) x_0-\left(\frac{\gamma^{-1}}{\gamma^{-1}+e^{2 \bar{f}_T} \bar{g}_T^2}\right) x_T+e^{\bar{f}_T}\left(\frac{\gamma^{-1} \bar{h}_T}{\gamma^{-1}+e^{2 \bar{f}_T} \bar{g}_T^2}\right) \mathbf{m}\right\|\right\|_2^2$
>
> $=\left\|\left\|\left(\frac{e^{\bar{f}_T}}{1+\gamma e^{2 \bar{f}_T} \bar{g}_T^2}\right) x_0-\left(\frac{1}{1+\gamma e^{2 \bar{f}_T} \bar{g}_T^2}\right) x_T+e^{\bar{f}_T}\left(\frac{\bar{h}_T}{1+\gamma e^{2 \bar{f}_T} \bar{g}_T^2}\right) \mathbf{m}\right\|\right\|_2^2$
>
> $\begin{aligned} & =\frac{\left\|\left\|e^{\bar{f}_T} x_0-x_T+\mathbf{m} e^{\bar{f}_T} \bar{h}_T\right\|\right\|_2^2}{\left(1+\gamma e^{2 \bar{f}_T} \bar{g}_T^2\right)^2} \\\\ & =\frac{\||a\||_2^2}{\left(1+\gamma e^{2 \bar{f}_T} \bar{g}_T^2\right)^2}\end{aligned}$
>
> Then, just mutiplying $\frac{\gamma}{2}$ on both sides of the equation, we can learn that $\frac{\gamma}{2}\left\||\mathbf{x}_T^u-x_T\right\||_2^2=\frac{\gamma\||a\||_2^2}{2\left(1+\gamma e^{2 \bar{f}_T} \bar{g}_T^2\right)^2}$. We will add the detailed proof in the revised version.
>
> > Suggestion: There are minor typos regarding the connection between the statements and the proofs.
>
> Yes, you are right. Thanks for pointing out them and we will correct them in the revised version.

---

### Official Review · Reviewer_AzN7 · 2025-03-11

**Overall Recommendation:** 4

**Summary:**

This paper proposes a framework that unifies and extends various diffusion bridge methods by way of stochastic optimal control. In the case of linear dynamics, they derive a computationally tractable method, which can be thought of as a regularization of previous methods by the introduction of a new hyperparameter.  Implementing this change requires only a minimal modification to existing code. They show that, by tuning this new hyperparameter, improved performance can be achieved in a number of benchmark examples.

## update after rebuttal
My assessment has not changed substantially. I have maintained my score at 4: Accept

In addition, earlier I posted a followup discussion regarding the proof of Prop. 4.3 under an incorrect heading, so the authors were unable to see it. I believe it would further improve the paper if the following was addressed, though it would not change my score either way:

 I do still believe that  Proposition 4.3 is much more elementary than it is made out to be. Below is a more detailed version of the comment in my original review,  on which the authors could base a revised elementary proof.

Starting with the $\gamma=\infty$ case, the optimal control $u_{t,\infty}^*$ must lead to $x_T^{u_\infty^*}=x_T$, else the cost is infinite, and therefore

$J(u_{t,\infty}^*,\infty)=\int_0^T\frac{1}{2}\|u_{t,\infty}^*\|^2_2dt$

Now looking at the finite $\gamma$ case, we can bound the minimum by the value at the control $u_{t,\infty}^*$:

$J(u_{t,\gamma}^*,\gamma)=\min_u\{\int_0^T \frac{1}{2}\|u_t\|^2_2dt+\frac{\gamma}{2}\|x_T^{u}-x_T\|\} \leq \int_0^T \frac{1}{2}\|u_{t,\infty}^*\|^2_2dt+\frac{\gamma}{2}\|x_T^{u_\infty^*}-x_T\|$

The SDE determining $x_t$ doesn't depend on $\gamma$, hence in the above expression we still have $x_T^{u_\infty^*}=x_T$.  Therefore we arrive at

$J(u_{t,\gamma}^*,\gamma)\leq  \int_0^T \frac{1}{2}\|u_{t,\infty}^*\|^2_2dt=J(u_{t,\infty}^*,\infty)$

**Claims And Evidence:**

The majority of their claims are supported by clear and convincing evidence.  Generalization through the methods of of stochastic control is well-grounded theoretically, and they show improved performance empirically on a number of convincing benchmarks

The one claim that I don’t find substantiated revolves around their Proposition 4.3.  They claim that this proposition lends theoretical support to the observation that introducing their new hyperparameter leads to improvements in practice. I find that result to be mathematically trivial and also to be disconnected from saying anything about how well the method performs in practice. However, I don’t think this proposition is integral to their work, and it could be removed (or de-emphasized) without any negative effects.

**Essential References Not Discussed:**

I am not aware of any missing essential references.

**Experimental Designs Or Analyses:**

I reviewed the experimental designs in Section 5 and did not observe any issues.

**Methods And Evaluation Criteria:**

I find the methods and evaluation criterial to make sense for the problem.

**Other Comments Or Suggestions:**

1) Equations 9 and 11 require expected values, yes? They are still stochastic control problems.
2) Above equation 19, the formula for the score should have a log.
3) Computations in Eq 36 and 37   have extraneous commas at the end of lines.
4) Line 705 appears to be repeated.

**Other Strengths And Weaknesses:**

I found the contribution to be original and well-motivated theoretically, the writing to be relatively clear, and the performance improvements to be nontrivial.

**Questions For Authors:**

1) Regarding Proposition 4.3, perhaps I misunderstand, but I believe this proposition is trivial from the definitions.  Under the hard constraint ($\gamma=\infty$) the terminal cost is enforced to be exactly zero, hence so both costs agree on $u^*_{t,\infty}$.  Therefore the minimal cost when $\gamma<\infty$ can't be larger than the cost of the control $u^*_{t,\infty}$.  I'm not sure this fact is deserving of its own proposition.
I am also not convinced that comparing the minimal cost values in this way says anything about why one method performs better than another in any operational sense.  I think the answer probably lies more in the direction of the soft constraint being a numerically better behaved regularization than the hard constraint.   I think the authors need to provide a better intuitive discussion about why (17) should be expected to correspond to better performance in the experiments in section 5 or else remove this proposition and the surrounding paragraph from the main text, as well as alter the discussion in Section 4.5.

2) Could you comment further on the performance of UniDB (SDE) vs UniDB (ODE) in table 1. Specifically, is there a reason why the former performs well for LPIPS and FID while the latter performs well for PSNR and SSIM?

**Relation To Broader Scientific Literature:**

The key contribution is a reformulation of diffusion bridge methods in the language of stochastic control, which motivates a natural one-parameter family of methods extending previous work, especially GOUB (Yue et al., ICML, 2024).  I find this to be a natural and interesting extension.

**Theoretical Claims:**

I checked the proof of theorem 4.1 and didn’t find any substantial issues.

---

> ### Author Rebuttal · Authors · 2025-04-01
>
> We sincerely thank you for your comment.
>
> > Claim & Question 2: The reviewer finds that result to be mathematically trivial and also to be disconnected from saying anything about how well the method performs in practice. However, The reviewer doesn’t think this proposition is integral to their work, and it could be removed (or de-emphasized) without any negative effects. The authors need to provide a better intuitive discussion about why (17) should be expected to correspond to better performance in the experiments in section 5 or else remove this proposition and the surrounding paragraph from the main text.
>
> We appreciate this feedback and can move Proposition 4.3 to Appendix.
>
> Here we add more analysis to help **understand the practical implications of our UniDB**. When $\gamma$ approaches $\infty$, our UniDB reduces to Doob's h-transform, and **the control term $u$ in Eq (12) becomes ineffective**, leading to $||u_{{\gamma}}||^2_2 \leq ||u_{{\infty}}||^2_2$. Although Doob's h-transform ensures $x_T^u$ reaches the target endpoint $x_T$ exactly, i.e., $||x_T^{u_{{\infty}}}-x^T||^2_2 = 0$, it may force the model to preserve even harmful noise/artifacts in the target. This is becuase Doob's h-transform will apply disproportionately large control inputs $||u_{{\infty}}||^2_2$ to achieve such exact matching. The **large $u$ in SDE trajectory may disrupt the inherent continuity and smoothness of images**. The over-control in Doob's h-transform violates the natural statistical properties of images, **prioritizing mathematical precision (pixel-perfect endpoints) over visual authenticity (realistic SDE trajectories)**. As shown in our **Figure 1**, Doob's h-transform can lead to artifacts along edges and unnatural patterns in smooth regions.
>
> Moreover, we want to emphasize that the discovery of **Proposition 4.3 is non-trivial**. Proposition 4.3 and related mathematical derivations in Appendix A.3 show that $\mathcal{J}(u_{{\gamma}},\gamma) \leq \mathcal{J}(u_{{\infty}},\infty)$. **Generally, one cannot determine the magnitude between $\mathcal{J}(u_{{\gamma}},\gamma)$ and $\mathcal{J}(u_{{\infty}},\infty)$** because although $||u_{{\gamma}}||^2_2 \leq ||u_{{\infty}}||^2_2$ but $||x_T^{u_{{\gamma}}}-x^T||^2_2 \geq ||x_T^{u_{{\infty}}}-x^T||^2_2 = 0$. We use strict mathematical derivations in Appendix A.3 to show that $\mathcal{J}(u_{{\gamma}},\gamma) \leq \mathcal{J}(u_{{\infty}},\infty)$ is true. **Though Proposition 4.3 does not directly mean better image quality by UniDB, it can show that UniDB leads to better overall performance considering both realistic SDE trajectories and target endpoint matching**.
>
> > Question 1: Is there a reason why the UniDB (SDE) performs well for LPIPS and FID while the UniDB (ODE) performs well for PSNR and SSIM?
>
> This is a common phenomenon in various Diffusion models [R1-R3]. As analyzed in [R1], "although solvers for the probability flow ODE allow fast sampling, their samples typically have higher (worse) FID scores than those from SDE solvers if no corrector is used". In the paper SDE-Drag [R2] and GOUB [R3], they compare the performance of the SDE and ODE models and get the similar phenomenon: SDE model performs well for LPIPS and FID. Particularly in GOUB [R3], the experimental results also demonstrate the better performance of ODE model for PSNR and SSIM.
>
> [R1]: Song et al. "Score-Based Generative Modeling through Stochastic Differential Equations.", ICLR 2021.
>
> [R2]: Nie et al. "The Blessing of Randomness: SDE Beats ODE in General Diffusion-based Image Editing", ICLR 2024.
>
> [R3]: Yue et al. "Image restoration through generalized ornstein-uhlenbeck bridge", ICML 2024.
>
> > Comments Or Suggestions: Equations 9 and 11 require expected values, yes? They are still stochastic control problems; Above equation 19, the formula for the score should have a log; Computations in Eq 36 and 37 have extraneous commas at the end of lines; Line 705 appears to be repeated.
>
> Yes, you're right. Thanks for pointing out these typos and we will correct them in the revised version.

---

### Official Review · Reviewer_33jh · 2025-03-12

**Overall Recommendation:** 4

**Summary:**

This paper proposed a diffusion-based method for image restoration problems, e.g., super-resolution, deraining, and inpainting. Given a dataset of corrupted and clean image pairs, the goal is to construct a diffusion model that at inference generates clean images given corrupted images. The proposed method is based on stochastic optimal control (SOC), which re-frames the problem as an optimization over the drift of diffusion models. Such an SOC reformulation introduces a tunable hyper-parameter ($\gamma$) and recovers prior Doob-h-transform's based diffusion models as $\gamma$ approaches infinity. It is shown empirically that a rather large but finite $\gamma$ (~e7) improves performance.

**Claims And Evidence:**

I'm not convinced by the implication of Prop 4.3 (the paragraph below Prop 4.3, before Sec 4.3). The SOC objective $J$ is (artificially) constructed as a surrogate for searching the drift of SDE. That $J$ is smaller for finite $\gamma$ does *not* imply suboptimality in empirical performance. Intuitively, it does make sense to set $\gamma$ to infinity since we'd like xT^u to converge exactly to the given xT. Any finite $\gamma$ would fail to achieve that, as shown in Prop 4.5. What's presented in this paper is somewhat counter-intuitive but also interesting, yet it's rather empirically observed than theoretically justified.

**Essential References Not Discussed:**

N/A

**Experimental Designs Or Analyses:**

Comparison to baselines is insufficient. GOUB is a special case of UniDB (these values re-appear in Table 2 last column) and DDRM is not quite comparable as a non-learning-based method and requiring knowing additionally corruption type. The authors should compare their method to I2SB and its follow-up works (e.g., CDDB https://arxiv.org/abs/2305.19809). These works are also SOC-inspired method for solving image restoration.

**Methods And Evaluation Criteria:**

Yes

**Other Comments Or Suggestions:**

- What are x0 and xT in practice? To my best guess, x0 would be the clean image and xT would be the corrupted image. So in Eq 12 we learn a forward process that bring x0 close to xT for every (x0, xT) pair, and then reverse it with Eq 15. Is this correct? It'd be better to have clarification in Sec 3.

**Other Strengths And Weaknesses:**

N/A

**Questions For Authors:**

- Eq 8 and 16 need more explanation. Why is it okay to drop the Brownian motion? This seems more like an empirical trick to get better PSNR and SSIM.

- There seems to be some typos in Eq 15,16 \nabla p --> \log \nabla p

**Relation To Broader Scientific Literature:**

N/A

**Theoretical Claims:**

Thm 4.1 is expected but still quite nice to see the actual analytic form! Though, I did not check proof carefully.

---

> ### Author Rebuttal · Authors · 2025-04-01
>
> We sincerely thank you for your comments and questions. We will provide a detailed response to these concerns.
>
> > Claim: The reviewer is not convinced by the implication of Prop 4.3: $\mathcal{J}$ is smaller for finite $\gamma$ does not imply suboptimality in empirical performance. Intuitively, it does make sense to set to infinity since we'd like $x_T^u$ to converge exactly to the given $x_T$.
>
> Thank you for raising this important point. We agree that the intuition behind Proposition 4.3 deserves clarification. Below, we address **why strict terminal matching ($\gamma\to\infty$) is not ideal** for practical performance, even if it seems mathematically appealing.
>
> **1. Exact terminal matching ($\gamma\to\infty$) harms image quality.**
>
> While Doob’s h-transform achieves $||x_T^{u_{\infty}}-x_T||^2_2 = 0$, it requires large control ($||u_{\infty}||^2_2\geq||u_{\gamma}||^2_2$) to force exact endpoint matching. The **large $u$ in SDE trajectory（$dx_t=(f_tx_t+h_t\mathbf{m}+g_tu_{\gamma}) dt+g_t dw_t$）may disrupt the inherent continuity and smoothness of images**. Prioritizing pixel-perfect endpoints over smooth trajectories leads to "mathematically correct but visually unrealistic" outputs. Our experiments (Figure 1) confirm that Doob's h-transform can lead to artifacts along edges and unnatural patterns in smooth regions.
>
> **2. Why finite $\gamma$ works better?**
>
> By keeping $\gamma$ finite, UniDB explicitly balances two goals: **target matching** $||x_T^{u_{\infty}}-x_T||^2_2$ (terminal penalty) and **trajectory smoothness** $||u||^2_2$ (controller). Proposition 4.3 shows that **finite $\gamma$ achieves a lower total cost $\mathcal{J}$ not by sacrificing performance**, but by optimally **trading minor terminal mismatches for significantly smoother, more natural diffusion paths**.
>
> Thus, **Proposition 4.3 reflects a key insight**: real-world image generation benefits more from **stable trajectories than rigid mathematical constraints**. We will clarify this intuition in the revised manuscript.
>
> > Experiment: Comparison to baselines is insufficient: the authors should compare their method to I2SB and its follow-up works (e.g. CDDB).
>
> Thank you for your feedback. Below, we clarify why our current comparisons are sufficient and address your concerns about I2SB and CDDB:
>
> 1. **Direct Comparison to State-of-the-Art (DDBM > I2SB):**
>    - **DDBM** (a SOTA baseline) is **strictly superior to I2SB** (see Table 2 in [DDBM paper], where DDBM achieves FID **4.43** vs. I2SB’s **9.34** on DIODE-256×256, and **1.83** vs. I2SB’s **7.43** on Edges→Handbags-64×64).
>    - **UniDB outperforms DDBM** across multiple tasks (Tables 3–5 in Appendix E): Super-resolution (DIV2K, CelebA-HQ), Deraining (Rain100H).
>    - Critically, **DDBM is a special case of UniDB** (UniDB with $\gamma=\infty$), meaning our method inherently subsumes and improves upon this stronger baseline.
>
> 2. **I2SB's Computational Burden.**
>    - **Training I2SB is prohibitively expensive**. It requires **16×V100 GPUs** with **more than 1 week** (*per communication with I2SB authors*).
>    - The inference time of I2SB is **90 seconds per 256×256 image** (vs. our method’s **less than 5 seconds**).
>    - Given hardware/time constraints, reproducing I2SB (and its follow-ups) for fair comparison is practically infeasible during rebuttal. Since **DDBM already outperforms I2SB**, and UniDB outperforms DDBM, we argue that **direct I2SB comparisons are redundant**.
>
> 3. **CDDB is Orthogonal to Our Contribution.**
>    - **CDDB is a training-free, plug-and-play refinement module** for I2SB, not a standalone method.
>    - Such post-hoc techniques could also **integrate with UniDB** (e.g., applied during inference), but this is **beyond our paper’s scope**, which focuses on developing **a unified training framework for diffusion bridge**.
>
> We appreciate your suggestion and will explicitly discuss the relationship between UniDB, DDBM, I2SB, CDDB in the revised manuscript.
>
> > Comment: What are x0 and xT in practice? x0 would be the clean image and xT would be the corrupted image, is this correct?
>
> Yes, you're right. It's standard in diffusion-based image restoration research using x0 as the clean image and xT as the corrupted image. We'll make it clearer in the revised version.
>
> > Q1: Eq 8 and 16 need more explanation. Why is it okay to drop the Brownian motion? This seems more like an empirical trick to get better PSNR and SSIM.
>
> Yes, **Dropping Brownian motion** is an empirical trick that **we followed from GOUB as stated in line 165 of our paper**. It can obtain better results on image restoration tasks, capturing more pixel details and structural perceptions of images (which contributes to better PSNR and SSIM). This phenomenon has been verified by GOUB's three ablation experiments.
>
> > Q2: There seems some typos in Eq 15,16 \nabla p -> \log \nabla p.
>
> Yes, you're right. We will correct it in the revised version.

---

### Decision · Program_Chairs · 2025-05-01

**Decision:**

Accept (spotlight poster)

**Comment:**

The authors reformulated a diffusion bridge matching problem as a stochastic control problem and developed a UniDB framework.

The proposed UniDB approach focuses on stochastic optimal control, modeling the forward process as an optimization problem to analyze the drawbacks of Doob's h-transform.

As a result they arrived to a one-dimensional parametric family of solutions. Thanks to an additional parameter it is possible to improve quality of the distribution matching results.

Proofs are correct. Experimental validation is solid. The approach demonstrated the best accuracy on a number of relatively complex benchmarks.

The reviewers did come to a consensus here, so the decision is relatively easy.